# Optimisation of a thermal desorption-gas chromatography-mass spectrometry method for the analysis of monoterpenes, sesquiterpenes and diterpenes

Aku Helin[1], Hannele Hakola[1], and Heidi Hellén[1]

[1]Atmospheric Composition Research, Finnish Meteorological Institute, Helsinki, FI-00101, Finland

*Correspondence to*: Aku Helin (aku.helin@fmi.fi)

**Abstract.** In this study, a thermal desorption-gas chromatography-mass spectrometry (TD-GC-MS) method following sorbent tube sampling was developed for the determination of monoterpenes (MTs), sesquiterpenes (SQTs) and diterpenes (DTs) in gas-phase samples. The analytical figures of merit were determined, and the method performance was tested by conducting

experiments related to, for example, sampling recovery, storage stability and ozone reactivity. The limit-of-quantification values were 13–518 pg (0.5–9.3 pptv), intermediate precision was in the range of 3–10 % and the expanded measurement uncertainty was in the range of 16–55 % for terpenes. The sampling recoveries of terpenes were approximately within 100±20 % with different inlet lines (15 m long Teflon and 1 m long heated stainless-steel) and branch enclosure cuvette (6 L Teflon bag) tested. Ozone is an important factor causing losses of the studied compounds during sampling. Therefore, losses of

terpenes upon ozone exposure were studied and the reaction rate coefficients were estimated. The ozone reaction rate coefficient ($k_{O3}$) of ent-kaurene was experimentally estimated to be 2 orders of magnitude greater than the respective literature $k_{O3}$ value, demonstrating the potential underestimation of DTs contribution to atmospheric reactivity. The preliminary comparison between offline and online mode TD-GC-MS sampling and analysis revealed that diterpenes and oxygenated sesquiterpenes are lost in excessive amounts in online mode sampling, hindering the online mode applicability for the

quantitative analysis of these compounds. A few applications to real samples were tested to identify DTs potentially emitted by boreal forest tree species. In dynamic headspace samples of pine needles and spruce twigs heated to 60 °C, five DTs and 13 DTs could be detected in emissions, respectively. The semi-quantitatively estimated emission rates of DTs were roughly 1 to 3 orders of magnitude lower than those of MTs and SQTs. Similarly, in spruce branch enclosure emissions from a living tree, six DTs were detected once the enclosure was heated to ca. 60 °C. In summary, the developed analytical procedure was

demonstrated to be applicable for the analysis of MTs, SQTs and DTs. In addition, DTs could be detected in needles, twigs and branch enclosure emissions, however, high temperatures were required to promote the emissions and for obtaining detectable concentrations.

## 1 Introduction

Biogenic volatile organic compounds (BVOCs) are directly emitted from the biosphere into the atmosphere. Globally, BVOC emissions are estimated to reach ca. 760 Tg (C) yr$^{-1}$ and chemically consist mainly of isoprene, monoterpenes (MTs, $C_{10}$), sesquiterpenes (SQTs, $C_{15}$), methanol and acetone (Guenther et al., 1995;Guenther et al., 2012;Sindelarova et al., 2014). Most of the emitted BVOCs have high reactivity in the atmosphere and their lifetimes vary from minutes to hours (Atkinson and Arey, 2003). Therefore, they strongly affect the oxidative capacity of the atmosphere both regionally and globally. In the reactions, BVOCs form less volatile compounds and participate in secondary organic aerosol (SOA) formation, thus also affecting Earth's radiative budget (Ehn et al., 2014;Hoffmann et al., 1997;Jimenez et al., 2009).

During the last decade, a new technique to measure atmospheric total hydroxyl (OH) radical reactivity has been introduced to study indirectly the volatile organic compound (VOC) content of the atmospheric air (Sinha et al., 2008; Yang et al., 2016). By measuring how much OH radicals are consumed in the reactions and by comparing this amount with the known amount of VOCs obtained from chemical composition measurements, it can be evaluated how much unknown reactive compounds there are in the atmosphere. Several studies have shown that the gap between known and unknown compounds can be fairly large (Yang et al., 2016), for example, in boreal forest the missing fraction is approximately 50 to 90 % (Nölscher et al., 2012;Praplan et al., 2019;Sinha et al., 2010). Therefore, there is a need to identify compounds responsible for this missing atmospheric OH reactivity.

Terpenoids, mostly isoprene and MTs, have been studied quite intensively during past decades in boreal forest areas (Eerdekens et al., 2009;Hakola et al., 1998, 2003, 2009;Kontkanen et al., 2016;Rinne et al., 2007;Vanhatalo et al., 2018). While there are also studies on SQTs (Hakola et al., 2001, 2006, 2017;Hellén et al., 2018;van Meeningen et al., 2017;Yassaa et al., 2012), no atmospheric observations of diterpenes (DTs, $C_{20}$) yet exist in the boreal forest areas. However, DTs have been found in essential oils of some boreal forest tree species (Judžentienė et al., 2006;Kanerva et al., 2008;Kupcinskiene et al., 2008;Tumen et al., 2010), for example, in Scots pine needles grown in northern Europe the DTs comprised 6–14 % of the essential oil content (Judzentiene and Kupcinskiene, 2008). Even though diterpenes have not been detected (or *specifically* studied) in boreal forest air, they have been recently observed in atmospheric air and branch enclosure emissions in other environments (Chan et al., 2016;Haberstroh et al., 2018;Li et al., 2019;Matsunaga et al., 2012;Yáñez-Serrano et al., 2018;Yee et al., 2018). For example, at a rural site in Brazil the mean concentrations of DTs in air varied between 10–86 ppqv (Yee et al., 2018), and in Mediterranean gum rockrose enclosure the emission rates of DTs were in the range of 8x10$^{-5}$–14x10$^{-5}$ µg g$^{-1}$ h$^{-1}$ (Haberstroh et al., 2018). Therefore, it could be that DTs are one of the compounds included in the missing OH reactivity group. In addition, DTs are potentially contributing in secondary organic aerosol formation (Yee et al., 2018).

Diterpenes have fairly low vapor pressures and they can be categorised as semi- or low-volatile organic compounds. The determination of DTs in atmospheric gas-phase samples can be challenging due to their low-volatility and high reactivity, which might cause sampling line losses and emphasize the need for high-sensitivity detection methods (Yáñez-Serrano et al., 2018;Yee et al., 2018). In previous atmospheric studies, DTs and their derivatives have been analysed by using either

conventional chromatographic techniques or online mass spectrometric techniques (such as proton-transfer-reaction mass spectrometry) (Chan et al., 2016;Haberstroh et al., 2018;Li et al., 2019;Matsunaga et al., 2012;Yáñez-Serrano et al., 2018;Yee et al., 2018). Many of those chromatographic studies targeting DTs used traditional sorbent tubes for sampling prior to thermal desorption (or solvent desorption)-gas chromatography-mass spectrometry (TD-GC-MS) analysis (Haberstroh et al., 2018;Matsunaga et al., 2012;Yáñez-Serrano et al., 2018). In general, previously offline (and online) TD-GC-MS technique has been mainly used for the analysis of MTs and SQTs (e.g. Hellén et al., 2018;Mermet et al., 2019; Pankow et al., 2012), but the above-mentioned branch enclosure and ambient air studies indicate that this analytical technique is also applicable for the analysis of DTs (Haberstroh et al., 2018;Matsunaga et al., 2012;Yáñez-Serrano et al., 2018). Unfortunately, most of those studies did not present any detailed method descriptive performance parameters for DTs (e.g. detection limits, reproducibility, selectivity or sampling recovery), making it difficult to critically evaluate the suitability of the analytical procedures for the analysis of low volatility DTs (Haberstroh et al., 2018;Matsunaga et al., 2012;Yáñez-Serrano et al., 2018). In general, sorbent tube sampling followed by TD-GC-MS analysis methods have been demonstrated to be suitable for the analysis of MTs, however, there is very little detailed information about the suitability for the analysis of SQTs (Bouvier-Brown et al., 2009;Helmig et al., 2004;Jones et al., 2014;Mermet et al., 2019;Pankow et al., 2012), and data related to DTs is currently lacking.

In this study, sorbent tube sampling followed by TD-GC-MS analysis for the determination of MTs, SQTs and especially DTs in gas-phase samples was developed and evaluated. The TD-GC-MS method was partially incorporated from our group's previous studies (see e.g. Hellén et al., 2018), and modified accordingly to fit the needs for the analysis of low-volatility diterpenes. In principle, similar analytical procedures have been used previously in both ambient air and enclosure emission studies for the analysis of relatively volatile and semi-volatile MTs and SQTs in multiple field campaigns (Aaltonen et al., 2011;Hellén et al., 2018;Joensuu et al., 2016;Mäki et al., 2017;Vanhatalo et al., 2018). In this work, we build on our previous knowledge of the TD-GC-MS technique and developed our core TD-GC-MS method further to allow the analysis of the fairly low-volatility DTs. In this study, the method development and applicability related experiments done in view for the analysis of DTs included e.g. desorption efficiency, sampling recovery, stability tests, comparison of online and offline sampling modes and ozone reactivity tests. The method analytical figures of merit were also determined, including the quantification limits, intermediate precisions and measurement uncertainties for the terpenes. In addition, qualitative and semi-quantitative experiments were conducted in laboratory and in field conditions to characterise the DTs potentially emitted by boreal forest tree species.

## 2 Experimental

### 2.1 Chemicals and materials

Detailed information on chemicals is presented in the Supplement Table S1. The standard analytes list is presented in Table 1. The analyte included nine MTs (both hydrocarbon and oxygenated MTs), six SQTs (both hydrocarbon and oxygenated SQTs),

three diterpenes (all hydrocarbon DTs) and three other compounds that were categorised here as BVOCs. Unfortunately, only three commercially available DTs were obtained for this study and only two of these were analytically valid in terms of purity and traceability (Tables 1 and S1). The current supply situation of DTs is not optimal for the purpose of comprehensive method development, however, together with the results of MTs and SQTs to compare to, valuable information was expected be obtained even with only few DTs included as analytes.

Primary standard mix solutions of the target compounds were prepared by weighting and diluting the pure compounds in methanol. The analyte concentrations in primary solution were in the range of 40–200 mg L$^{-1}$ depending on the compound. Six calibration standard solutions were prepared by pipetting appropriate amount of primary solution and diluting with methanol. The standard solution concentrations ranged from 40–200 µg L$^{-1}$ (lowest standard) to 8000–40000 µg L$^{-1}$ (highest standard). The primary and standard solutions were stored in the dark at 4 °C.

The stainless-steel multiphase adsorbent (Tenax TA/Carbopack B) tubes (o.d. 1/4 in. x 3½ in.) used in this study were either purchased directly from PerkinElmer Inc. (Waltham, MA, USA) or prepared manually in the laboratory. The in-house packed sorbent tubes were prepared by packing empty stainless-steel tubes (PerkinElmer Inc., Waltham, MA, USA) with Tenax TA (60-80 mesh) and Carbopack B (60-80 mesh), both materials purchased from Sigma-Aldrich (St. Loius, MO, USA). Silanized glass wool (Phase Separations Ltd., Deeside, UK), stainless-steel mesh (Markes International, Llantrisant, UK) and gauze-retaining spring (Markes International, Llantrisant, UK) were used to prevent sorbent phase mixing and exiting. The adsorbent tubes were preconditioned at 300 °C for 150 min under helium flow. Prior to each use, the sorbent tubes were always conditioned for a minimum of 10 min at 300 °C. With these sorbent tubes, the sampling flow rate used throughout this study was typically 80–100 mL min$^{-1}$. The commercial and in-house prepared tubes were used collectively and randomly in this work.

The sorbent tubes were also used for TD-GC-MS calibration. Calibration tubes were prepared by injecting 5 µL of individual standard solution into the tube and simultaneously applying 80 mL min$^{-1}$ flow of nitrogen (≥99.9999 % AGA, Espoo, Finland). The tubes were purged for 10 min in order to evaporate the excess methanol (Hakola et al., 2003).The calibration concentration ranges were 0.2–55.8 ng for MTs, 0.2–61.2 ng for SQTs, 0.4–200 ng for DTs and 0.2–43.2 ng for other BVOCs (expressed as absolute amount (ng) in sorbent tube).

## 2.2 Thermal desorption-gas chromatography-mass spectrometry

As mentioned before in the Introduction, the TD-GC-MS method used here was partially applied from a previously developed and validated in-house method that has been used in our group's previous studies for the analysis of MTs and SQTs (see e.g. Hellén et al., 2012, 2018). In this current work, the analytical GC column stationary phase (nonpolar 1,4-bis(dimethylsiloxy)phenylene polydimethylsiloxane) film thickness was decreased from a previously used thick phase 1.0 µm column to 0.25 µm column as a compromise to maintain proper selectivity between the terpenes and to elute the DTs in reasonable time (chromatograms presented later on). It was expected that the DTs might retain too strongly in the 1.0 µm film thickness column and that the column bleed could become a problem in long-term use since high temperatures are required to

elute the DTs. Thus, the analytical column was replaced, GC oven temperature program was modified, and the MS scan settings were updated and modified. For the most part, the TD unit parameters' were kept as in the previously optimised in-house method, however, the desorption efficiency was separately tested and verified in this work.

The analyses were performed by using two different thermal desorption-gas chromatography-mass spectrometers (TD-GC-MSs), which are identified and abbreviated hereafter as TD-GC-MS1 and TD-GC-MS2. Most of the analytical method development and performance tests were conducted with TD-GC-MS1, whereas TD-GC-MS2 was used mainly for real sample applications. The details of both instruments and methods are presented in Table S2 and mentioned here briefly. The TD-GC-MS1 consisted of an automatic TD unit (TurboMatrix 350) connected to a GC (Clarus 680) coupled to a quadrupole MS (Clarus SQ 8 T), all units purchased from PerkinElmer Inc. (Waltham, MA, USA). The cold trap in TD and column in GC were Tenax TA/Carbopack B and Elite-5MS (60 m x 0.25 mm (i.d.), film thickness 0.25 µm), respectively (both from PerkinElmer Inc., Waltham, MA, USA). The TD-GC-MS2 consisted of an automatic TD unit (TurboMatrix 650) connected to a GC (Clarus 600) coupled to a quadrupole MS (Clarus 600 T), all purchased from PerkinElmer Inc. (Waltham, MA, USA). In this TD-GC-MS2, the cold trap in TD and column in GC were Tenax TA (PerkinElmer Inc., Waltham, MA, USA) and DB-5MS (50 m x 0.25 mm (i.d.), film thickness 0.25 µm, from Agilent Technologies, Palo Alto, CA, USA), respectively. The main differences in these TD-GC-MSs were the models of units, different cold traps and columns (same stationary phases, but 10 m shorter column in TD-GC-MS2) installed. Also, the TD-GC-MS1 has an online sampling feature in the TD unit (discussed later on), whereas the TD-GC-MS2 can be used only for offline sorbent tube analysis.

With both instruments, the optimised offline TD-GC-MS analysis was performed fundamentally in a similar manner, although the duration of different steps differed slightly (for details, see Table S2). Briefly, the sorbent tube was first dry purged in the TD for either 1 min or 5 min at room temperature at a flow rate of 50 mL min$^{-1}$ of helium ($\geq$ 99.9996 %, AGA, Espoo, Finland). In the primary desorption, the sorbent tube was desorbed for 5 min at 300 °C at a flow rate of 50 mL min$^{-1}$ of helium. The desorbed compounds were trapped into a cold trap held at 20 °C during the primary desorption. Then, in the secondary desorption, the cold trap was rapidly heated to 300 °C and held for either 1 min or 5 min. The helium flow rate during the cold trap desorption was either 10 or 30 mL min$^{-1}$, of which 1 mL min$^{-1}$ was entering the GC column and rest was passed to outlet split. The temperature of the heated valve in the TD was 220 °C. The heated line temperature between TD and GC was set to 200 °C. With TD-GC-MS1, the oven temperature program was as follows: 60 °C (held for 2 min), then to 300 °C at 8 °C min$^{-1}$ and 300 °C (held for 15 min). With the TD-GC-MS2, the oven temperature program was otherwise similar as with TD-GC-MS1, except the final step 300 °C was held only for 3 min. The flow rate of helium carrier gas into the analytical column was 1.0 mL min$^{-1}$ with both instruments. The GC-MS transfer line temperature was 220 °C with both TD-GC-MSs. In MS, electron ionisation at 70 eV was used. Total ion chromatogram (TIC) was scanned in m/z range 50–350 and selected ion recording (SIR) time windows with quantifier and qualifier ions were applied for the target analytes. The selected quantify m/z ions were typically base peaks or molecular ion peaks (Table 1).

### 2.3 Method development and experiments

#### 2.3.1 Desorption efficiency

Desorption efficiency was tested by doing two consecutive desorptions from the same sorbent tube loaded with analytes (c=40–200 ng). Triplicate measurements were done. Desorption efficiency was calculated by dividing the amount detected in the first desorption by the total amount detected in both desorption cycles.

#### 2.3.2 Stability in storage

The stability of the analytes in sorbent tube (c=40–200 ng) was evaluated by storing the tubes (*n*=2-6) sealed with brass
Swagelok caps and PTFE ferrules at 4 °C in the dark for ca. 1 week (5 days), 1 month (32–33 days) and 2 months (62 days). The stability after storage was compared to sorbent tubes analysed immediately after standard preparation.

#### 2.3.3 Sampling recovery

The inlet sampling line recovery tests were done by using two different commonly used inlets and at two different relative humidity (RH) levels. The inlet recovery experiments were done in a similar manner as in Hellén et al. (2012). The schematic
illustrations of the setups used in the experiments are presented in Fig. S1. Briefly, analyte solution was injected into a zero air stream via a heated (ca. 60±5 °C) Teflon PTFE t-piece by using an automatic syringe pump (at injection flow rate of 15 µL h$^{-1}$). The zero air was produced either by using a zero air generator (HPZA-7000, Parker Balston, Lancaster, NY, USA) or by passing laboratory room air through a carbon cartridge (100 g active carbon from Pall Life Sciences, Ann Arbor, MI, USA). The main flow rate was 0.8–1.0 L min$^{-1}$. The volatilised analytes passed through a 4 m long Teflon fluorinated ethylene
propylene (FEP) tubing (i.d. 1/8 in.), which was used as a mixing line, after which the first sorbent tube sample was taken. The second sorbent tube sample was taken after the inlet line under investigation. Inlet line recovery was calculated simply by the relationship of analytes found in the second tube divided by the amounts in the first tube.

The inlets tested were a 15 m long Teflon FEP tubing (i.d. 1/8 in.) and a 1 m long heated stainless-steel (grade 304) tube (i.d. 1/16 in., heated to ca. 150 °C), of which the latter is normally used for ozone removal purposes (Hellén et al., 2012). Both inlet
lines tested here are typically used in BVOC studies (Hakola et al., 2017;Hellén et al., 2018). The RH levels were attempted to control to close to 0 % and 100 % to cover the extremes, but in reality these were roughly 13±5 % (RH=0 %) and 87±6 % (RH=100 %). The RH was measured with a Vaisala HMI 33 device (probe HMP 35, Vaisala, Helsinki, Finland). All experiments were performed at room temperature (T=22±1 °C).

In addition to inlet sampling line recoveries, an enclosure cuvette recovery was tentatively tested. The cuvette was a ca. 6.2 L
bag made of Teflon FEP (wall thickness 50 µm); similar cuvette has been used previously in branch enclosure studies (Hakola et al., 2017). The cuvette recovery was tested in a similar manner as the inlet line recovery: the first sorbent tube was placed at the inlet end of the cuvette and the second tube at the outlet end of the cuvette (Fig. S1c). Three different flow rate

combinations were tested: 1.0, 2.0 and 6.7 L min$^{-1}$ at the inlet end and 0.7, 1.7 and 6.4 L min$^{-1}$ at the outlet end of the cuvette, respectively. The cuvette bag was not air-tight, so excess flow leaked out freely.

### 2.3.4 Comparison between online and offline mode sampling

As mentioned earlier, the TD-GC-MS1 has an online sampling feature in the TD unit. In online sampling mode, the air sample is drawn through an empty tube onto the cold trap where the analytes are trapped and preconcentrated. This online mode sampling feature has been used previously in field studies for the analysis of MTs and SQTs, e.g. in Hellén et al. (2018). The online sampling mode was compared to offline sampling mode with the TD-GC-MS1. Known amount of standard solution was injected into a 1.0 L min$^{-1}$ flow rate of zero air stream. After a ca. 4 m long Teflon FEP mixing line, both the offline sorbent tube sample (at 100 mL min$^{-1}$) and the online sample (at 40 mL min$^{-1}$) were taken concurrently (for 30 min) as illustrated in Fig. S2. A more detailed explanation of the experiments are presented later on.

### 2.3.5 Analytical figures of merit

The limit of quantification (LOQ) values for the TD-GC-MS method were determined by analysing multiple blank sorbent tubes (*n*=10). The LOQ was calculated following Eq. (1):

$$LOQ = A + 10 \times SD \,, \tag{1}$$

where A=average peak area and SD=standard deviation. Repeatability was calculated by analysing standard samples (*n*=6) during one day, and intermediate precision (or *reproducibility within-laboratory*) was obtained from analysing standard samples (*n*=22) during 4.5 months period (c=10-50 ng in sorbent tube). The expanded measurement uncertainty (U) was estimated from partial uncertainties by following ACTRIS (Aerosol Clouds Trace gases Research InfraStructure) guidelines (ACTRIS, 2018). The U (%) was estimated for two analytical procedures: i) offline sorbent tube sampling of ambient air followed by TD-GC-MS analysis and ii) offline sorbent tube sampling of emissions from a branch enclosure cuvette followed by TD-GC-MS analysis. The intermediate precision and recovery experiment results were included in the calculations of U. In the ambient air sampling procedure, the ozone removal inlet recovery results were used in the calculations, whereas in the branch enclosure emissions procedure the cuvette recovery results were used in the calculations. A more detailed description of the uncertainty calculations can be found in Supplement Text S1.

### 2.3.6 Ozone reactivity tests

It is well-known that compounds can be lost during sampling due to reactions with ozone. Since some terpenes are highly reactive towards ozone and the ozone reaction rate coefficients ($k_{O3}$) of diterpenes are uncertain (Haberstroh et al., 2018;Yáñez-Serrano et al., 2018;Yee et al., 2018), experiments focusing on analyte stability upon ozone exposure were conducted. The sorbent tube was loaded with known amount of analytes (c=40–200 ng) and the tube was flushed with either 0 ppb or 40 ppb of ozone during different times (from 0 to 240 min at 100 mL min$^{-1}$ flow rate). The total purge volume of the sorbent tube was

0–24 L. The ozone concentration of 40 ppb was selected since it represents a typical upper concentration level in background air in southern Finland (Anttila and Tuovinen, 2010). Ozone was produced by using an ozone generator (model 49C $O_3$ Calibrator, Thermo Environmental Instruments Inc., Franklin, MA, USA). The schematic illustration of the setup is presented in Fig. S3.

## 2.4 Applications to real samples

Two types of real sample applications were selected in order to test the analytical method performance, to identify compounds emitted by boreal forest tree species and to quantify the emissions from a branch enclosure. The first sample application was a dynamic headspace sampling of compounds emitted by needles/twigs in a closed chamber upon heating, and the second sample application was a study of emissions from a branch enclosure in field conditions. The main emphasis on real sample applications was set on the identification and study of DTs, although MTs and SQTs were also monitored. Based on previous studies, it was considered that since the DTs seem to be emitted from plant storage pools and since temperature has been shown to be a driving force for DTs emissions (Haberstroh et al., 2018;Matsunaga et al., 2012;Yáñez-Serrano et al., 2018), heating of the plant materials would be an efficient way for releasing the DTs into gas-phase. Thus, the studied plant material samples were firstly heated to temperatures ranging from 30 °C to up to 70 °C in order to obtain presumably detectable amounts of DTs to allow their identification and quantification.

Pine (*Pinus sylvestris*) needles and spruce (*Picea abies*) twigs were collected during early September 2018 in a boreal forest at the SMEAR II station (Station for Measuring Forest Ecosystem-Atmosphere Relations) in Hyytiälä, Finland (Hari and Kulmala, 2005). These plant material samples were qualitatively and semi-quantitatively analysed for diterpene content. The samples were stored in the dark at -18 °C for a maximum of 2 months prior to analysis. In a set of experiments, approximately 5–7 g of frozen pine needles or small pieced spruce twigs were placed inside a screw-capped 500 mL Duran bottle. The bottle was equilibrated at room temperature for 30 min prior to starting headspace purge-and-trap extraction experiments. The flow in and out of the chamber were set to 200 mL min$^{-1}$ and 100 mL min$^{-1}$, respectively. During the dynamic sampling, the bottle was placed inside an oven and heated in 5–10 °C steps from room temperature to up to 70 °C. The sorbent tube samples were collected outside the oven at room temperature for 20 min (sampling volume 2 L). The schematic illustration of the setup is presented in Fig. S4. The dry weight of the sample needles were 1.66–2.70 g (drying at 70 °C for 24–48 h). These samples were analysed with emphasis solely on the identification of DTs. Blank samples (empty chamber heated) were collected prior to experiments.

Branch enclosure emissions were studied on-site at the SMEAR II station as a proof-of-concept type of experiment on two different days (i.e. to test and to demonstrate that the analytical procedure is suitable for the purpose it was intended for). Spruce (*Picea abies*) branch with needles was placed inside a ca. 1.2 L Teflon bag cuvette, which was connected to a fixed plate with inlet and outlet ports. The incoming air (at flow rate of 8.2 L min$^{-1}$) was purified with a high capacity gas purifier (Supelco, Bellefonte, PA, USA) and passed through a ca. 1 m long copper tube (o.d. 12 mm) wrapped with a heating wire and insulator cover (copper tubing was used for improving heat transfer). The incoming air was heated to increase the cuvette

inside temperature in order to promote the emissions of DTs. A data logger with temperature sensor was used for monitoring the cuvette inside temperature. The cuvette was additionally covered with aluminium foil for further insulation in order to achieve sufficiently high temperatures. The cuvette inside temperature ranged from 14 to 61 °C and the heating gradients are presented in Fig. S5. Sorbent tube samples were collected either directly from the outlet port of the cuvette or from a bypass flow (outlet flow rate 1.0 L min⁻¹). The schematic illustration of the setup is presented in Fig. S6. The branch enclosure experiment sampling dates were 6[th] and 10[th] of August, 2019. On the former date, the samples were collected from a bypass flow and on the latter day directly from the cuvette outlet port. The branch was cut immediately after the sampling was completed, thus, it is plausible that the tree was under stress on the latter day experiments. The dry weight of the needles were 6.35–6.75 g (drying at 70 °C for 24–48 h). Blank samples (empty cuvette heated) were collected prior to experiment.

## 2.5 Chromatographic data analysis

The non-target compounds in real sample applications were tentatively identified based on both retention index (RI) values and mass spectra comparison to NIST mass spectral library and/or Adams (2007) library. The n-alkane based RI values were calculated based on both Kovats Index (KI) and Arithmetic Index (AI) as shown in Adams (2007). If an unknown compound could not be tentatively identified (either too high RI difference or not adequately matching mass spectra), a proposed compound formula was deduced based on the mass spectra. The non-target compounds that were either tentatively identified or assigned with a formula were then categorised as MTs, SQT and DTs when applicable, the others were categorised as BVOCs. For example, if the non-target compound had the characteristic m/z 272, 257, 243 and 229 ions of DTs and the RI value was representative of DTs (Adams, 2007), the compound was assigned in this study to the plausible DT category. Similar identification procedure of non-target compounds has been used also in previous BVOC studies (e.g. Chan et al., 2016; Hellén et al., 2018; Kännaste et al., 2013; Yee et al., 2018). The compounds were marked here as *tentatively identified* since authentic standards were not available, thus the final verification is lacking. The analysed blank samples were used to exclude possible contaminant peaks.

The non-target compounds detected in real samples were quantified by using a compound from the standard analytes list (α-pinene, limonene or linalool were used for MTs and for close eluting BVOCs, β-caryophyllene or caryophyllene oxide for SQTs and for close eluting BVOCs, and cembrene or ent-kaurene were used for DTs and close eluting BVOCs). Since the non-target compounds were quantified in this manner, the mass concentration results are merely semi-quantitative. A signal-to-noise (S/N) ratio of 10 criteria was used in non-target compound quantification. Similar quantification procedure for non-target compounds has been used also in previous BVOC studies (e.g. Bouvier-Brown et al., 2009; Haberstroh et al., 2018; Hellén et al., 2018).

Emission rate (E) was calculated by using Eq. (2):

$$E = \frac{(C_{out} - C_{in}) \times F}{m},$$
(2)

where $C_{out}$ is the concentration in the outlet flow, $C_{in}$ is the concentration in the inlet flow, F is the flow rate into the cuvette and m is the dry weight of the needles (Ortega and Helmig, 2008). In both real sample applications, the $C_{in}$ was negligible (or <LOQ) for all compounds, and was subsequently set to zero in the calculations. No blank subtraction was needed in the calculations since most analytes were absent in blank samples.

# 3 Results and discussions

The first part of this section covers the TD-GC-MS performance evaluation experiments, and in the latter part the qualitative and semi-quantitative results from the real sample applications are presented.

## 3.1 Method performance results

### 3.1.1 Desorption and chromatographic separation

The desorption efficiency was ≥99.8 % for all compounds with the TD-GC-MS1 at 300 °C for 5 min. With the TD-GC-MS2 desorption time needed to be increased from 1 min to 5 min at 300 °C (both sorbent and cold trap) to minimise the carryover, which arise either from the sorbent tube or cold trap, and to obtain sufficient desorption efficiency (≥99.7%). Higher desorption temperatures were not tested to avoid sorbent deterioration in long-term use.

The chromatogram from a standard solution analysis with TD-GC-MS1 is shown in Fig. 1, and more detailed versions are shown in Figs. S7-8 for both TD-GC-MS instruments. The separation was considered to be adequate since all compounds could be baseline separated. The RI values for each compound are presented in Table 1 (according to run with TD-GC-MS1).

### 3.1.2 Compound stability in storage

The stability was acceptable (>80 %) with most of the compounds still after the 2 months storage period (Fig. S9). On average, MTs were recovered at 101±2 %, 93±5 % and 97±4 % after 5 days, 1 month and 2 month storage, respectively. Similarly, SQTs were recovered on average at 104±2 %, 89±3 % and 94±5 % after 5 days, 1 month and 2 month storage, respectively. Of the diterpenes, ent-kaurene and 3-MA were recovered even after the 2 months storage within acceptable limits (roughly 80±25 %, Fig. S9), however, cembrene had been lost to 49±18 % already after 1 month storage. It is unknown why cembrene seem to have decomposed during the first month of storage, yet then stayed relatively stable until the next month (Fig. S9, from 49±18 % to 52±33 %). No decomposition product of cembrene could be identified in the chromatograms.

### 3.1.3 Recovery results

In Fig. 2a are shown the amount of each compound detected in the first sorbent tube (after the mixing line, Fig. S1a) and the expected theoretical amount calculated based on the experimental conditions used. As can be seen in Fig. 2a, the observed amounts were mostly in close agreement with the expected amounts. Although this was not a prerequisite for the recovery

experiments, the relative recovery was calculated based on amounts found in tubes before and after the sampling line, it demonstrates that the experimental setup was working adequately in practice (although the reproducibility was a bit poor for some compounds, see Fig. 2a). The injection t-piece needed to be heated to 60 °C in order to increase the volatilisation of the diterpenes. If there was no heating applied, then e.g. ent-kaurene yield in the first tube was less than 20 % of the expected amount (data not shown). The diterpenes were observed to stick into the walls of the injection PTFE t-piece if it was not heated, thus heating is recommended to be used in similar experiments. Higher temperatures were not tested here in order to avoid the possible PTFE contamination peaks.

In Fig. 2b are shown the inlet recovery results for both Teflon FEP and ozone removal inlet lines. The recovery results were in acceptable level (100±20 %) for most of the compounds with both inlets. Interestingly, even though heating was required for volatilisation at the injection point, DTs passed the sampling lines completely in gas-phase as demonstrated by the acceptable recoveries. This could have been expected in the case of the heated stainless-steel line (which certainly maintains the compounds in gas-phase), but especially the recoveries obtained with the 15 m long Teflon line were very promising in view of real applications. The results indicate that under the applied conditions the Teflon FEP sampling lines do not need to be heated excessively above room temperature (22±1 °C).

In Figs. 2c-2d are shown the recovery values obtained with both inlets at the different RH levels tested. The RH did not affect the recovery levels dramatically with either inlet tested. Only 3-MA had a slightly worse recovery (78±12 %) when Teflon inlet was used at low RH level, but acceptable when the RH was high (recovery 89±6 %). The standard deviation was fairly large for some of the least volatile compounds (ent-kaurene, cembrene, 3-MA and caryophyllene oxide) and occasionally for β-farnesene and terpinolene, which might be due to a combination of the compound properties and experimental variation.

The cuvette recovery results showed some variability (Fig. 3). As mentioned in Sect. 2.3.3, three different flow rates were tested (inlet flows 6.7, 2.0 and 1.0 L min$^{-1}$). The recovery values were acceptable at all flow rates tested for all other compounds except for the diterpenes (Fig. 3). The recoveries of MTs and SQTs were on average 95±1 % and 94±2 % at 1.0 L min$^{-1}$ and 96±1 % and 93±3 % at 6.7 L min$^{-1}$ flow rates, respectively. However, the recoveries of DTs were on average 44±11 %, 80±6 % and 76±4 % at flow rates of 1.0, 2.0 and 6.7 L min$^{-1}$. The poor recoveries of DTs observed at the lowest flow rate tested indicate that the residence time inside the cuvette was likely too long for these low volatility compounds (note that for MTs and SQTs the recoveries were acceptable). The DT recoveries were seemingly adequate with the other flow rates tested, indicating that the set flow rate range of ca. 2–7 L min$^{-1}$ would be suitable for sampling. However, with the highest flow rate tested, the absolute mass concentrations of DTs were very close to LOQ values (Fig. 3). This was likely related to an experimental problem in the analyte volatilisation setup. The injection t-piece heating was not efficient enough for allowing complete volatilisation of the DTs, since the high flow rate rapidly cooled the t-piece. Therefore, these DT cuvette recoveries need to be considered as tentative results, since the concentrations levels were not systematic between the different flow rates tested (Fig. 3). A better sample introduction system would be needed to get the volatilisation more systematic at all flow rates under investigation.

### 3.1.4 Analytical figures of merit of the developed TD-GC-MS method

The analytical figures of merit are shown in Table 1. The within-day repeatability ($n$=6) was fairly good for all compounds (relative standard deviation, RSD<5 %), except for caryophyllene oxide which had a fairly high variation (≈19 %). The intermediate precision ($n$=22 during 4.5 months period) was acceptable for all compounds (RSD<10 %), when calculated based on the quantified mass concentrations in the sorbent tube (Table 1). The expanded measurement uncertainty (U) at 95 % confidence level for terpenes varied between 16–32 % and between 18-55% with the ambient air sampling and branch enclosure emission procedures, respectively.

In general, the LOQ values (expressed as absolute amount in sorbent tube) were the lowest for monoterpenes (13–88 pg) and for sesquiterpenes (33–198 pg), and the highest for caryophyllene oxide (340 pg) and diterpenes (287–518 pg). The highest LOQ values were observed in the case of DTs likely because the baseline was the noisiest at the end of the chromatogram and because these compounds had minor contaminant peaks in some of the blank sorbent tubes. When converted to pptv units by using a sampling volume of 5 L, the LOQ values were approximately 0.5–2.8 pptv for MTs, 0.8–7.5 pptv for SQTs and 5.1–9.3 pptv for DTs. These are on a similar level as the method detection limits reported for MTs and SQTs in previous GC-MS studies (Bouvier-Brown et al., 2009;Hellén et al., 2018;Jones et al., 2014;Mermet et al., 2019;Sheu et al., 2018), e.g. Pankow et al. (2012) reported detection limits (S/N-ratio of 10) of 0.7–2.1 pptv for MTs and 0.9–1.4 pptv for SQTs. In addition, the LOQ values obtained in this study by TD-GC-MS were better than the detection limits recently reported by proton-transfer-reaction time-of-flight mass spectrometer (PTR-TOF-MS), which were approximately 90 pptv for MTs, 200 pptv for SQTs and 510 pptv for DTs (Yáñez-Serrano et al., 2018).

The calibration curve ranges were approximately 0.2–55.8 ng for MTs, 0.2–61.2 ng for SQTs and 0.4–200.0 ng for DTs. A fairly high concentration range was used in order to cover both the low (atmospheric air) and high (enclosure emissions) concentration applications with a single method. A quadratic fit and intercept set to zero was used in calibration for all compounds. The regression fit of the calibration curves was evaluated based on residual analysis. The calibration curve correlation coefficient was typically ≥0.998 for all compounds.

### 3.1.5 Online *vs.* offline sampling with TD-GC-MS1

A detailed version of the results from the comparison between offline and online mode sampling with TD-GC-MS1 is presented in the Text S2 (Figs. S10-S16). Here, a short summary of the observations is presented. In online sampling mode, samples are collected through the TD unit sampling lines, valves and an empty sorbent tube directly onto the cold trap (Fig. S10).

Collectively, the results indicated that some of the analytes are lost in excessive amounts in the online sampling mode. For example, when the material of the empty tube used in online sampling was stainless-steel, the concentrations of β-farnesene, caryophyllene oxide and DTs were below LOQ in online mode results. Yet, the recoveries ($c_{online}$/$c_{offline}$*100%) of MTs and other sesquiterpenes were acceptable, on average 96±6 % and 84±10 %, respectively. Interestingly, a substantial portion of the compound losses could be explained by the losses onto the empty tube used in online sampling, e.g. the amount of β-farnesene

lost in the empty tube was ca. 44%. However, simultaneously a substantial portion of the losses could not be explained by losses onto the empty tube, e.g. up to 98% of the ent-kaurene losses could not be accounted for. Somewhat similar observations have been made in previous studies, e.g. Arnts (2010) observed significant SQT losses on stainless-steel tube (about 60 % was missing) and Helmig et al. (2003) found that high sampling system temperatures (110–170 °C) were needed to prevent SQT losses in their setup. Heating to such high temperatures was not feasible in our setup and was not tested.

The online mode sampling efficiency of β-farnesene, caryophyllene oxide and DTs could be increased by changing the empty tube material from stainless-steel to glass and by adjusting the sample air relative humidity (Fig. 4). Increasing the sample air RH improved the recoveries of the aforementioned compounds no matter the tube material, however, the absolute recoveries were otherwise better with the glass tube than with the stainless-steel tube. Most notably, β-farnesene results were consistent and the recovery was good (92±5 %) when the glass tube was used in online mode. Caryophyllene oxide and diterpenes could be detected, but recoveries were fairly poor and had a high uncertainty.

In summary, the online sampling results were in close agreement with the offline sampling results for most of the analytes, demonstrating the online TD-GC-MS method applicability for the quantitative analysis of MTs and SQTs. However, a large discrepancy and poor repeatability was observed between online and offline mode sampling results in the case of diterpenes and caryophyllene oxide. By using glass tube (or glass coated stainless-steel tube, Text S2) and humidified air the recoveries improved, but further development (e.g. heating of the all lines, valves and sorbent tube in the TD) would be needed for quantitative analysis of diterpenes and caryophyllene oxide. Since the RH was shown to have an effect on the recoveries, humidity monitoring of the sample air stream is recommended

### 3.1.6 Ozone exposure and losses of terpenes

As explained in Sect. 2.3.6, sorbent tubes containing the analytes were purged with either 0 ppb or 40 ppb of ozone from 0 to 24 L of total purge volume. In theory, the absolute amount of a compound should stay constant in the sorbent tube if the compound is not reacting with ozone or if breakthrough volume is not reached. The zero experiments ($O_3$=0 ppb) were used as a reference to estimate the possible breakthrough, although it was not expected that the terpenes amounts would decrease dramatically due to breakthrough (Sheu et al., 2018). The ozone exposure experiments ($O_3$=40 ppb) were done in order to estimate the possible losses caused by ozone, and the results were also applied to estimate ozone reaction rate coefficients ($k_{O3}$) for the analyte diterpenes. The results for all compounds are shown in the Fig. S18 and few selected plots are presented in Fig. 5.

Based on the $O_3$=0 ppb purging results, breakthrough volume (50 % lost, $BTV_{50\%}$) was not reached with any of the compounds under the applied experimental conditions (Fig. S17). At the highest purge volume of 24 L, the relative amount of all compounds were higher than 0.87 and on average 0.97±0.05 for MTs, 0.96±0.05 for SQTs and 0.98±0.02 for DTs. Thus, any decrease observed in the following $O_3$=40 ppb results was attributed to be due to ozone reaction losses.

The $O_3$=40 ppb purging results seemed reasonable for most of the compounds when considering the compound specific $k_{O3}$ values (Fig. S17 and Table S3). In general, the linear regression fit (relative amount in sorbent as a function of purge volume)

slopes were declining the fastest with compounds that had the largest $k_{O3}$ values (i.e. compounds which react the fastest with ozone are subsequently lost rapidly from the sorbent tube). However, this trend analysis also revealed that the literature reported predicted diterpenes $k_{O3}$ values might be slightly misleading (Haberstroh et al., 2018). For example, as can be seen in Fig. 5 upper panel plots, the slope declined the fastest as the $k_{O3}$ increased in the order of camphene, limonene and β-caryophyllene, however, ent-kaurene $k_{O3}$ value is somewhat lower than would be expected based on the slope decline.

In the lower panel plot of Fig. 5, literature retrieved ln ($k_{O3}$) values were plotted as a function of the slopes obtained from the ozone purging experiments (Table S3 and Fig. S18), similar as done in Pollmann et al. (2005). As can be seen in Fig. 5 and Table S3, the predicted literature $k_{O3}$ value of cembrene is in close agreement with the obtained experimental value, whereas the corresponding ent-kaurene $k_{O3}$ value would be actually 2 orders of magnitude higher based on the linear regression fit. Based on this simple estimation process, the ozone reaction rate coefficients for ent-kaurene and cembrene were $(1.4\pm8.2)\times10^{-15}$ and $(2.4\pm15.5)\times10^{-15}$ cm$^3$ molecule$^{-1}$ s$^{-1}$, respectively. Consequently, the estimated lifetimes of DTs were on average $11\pm4$ min, whereas the estimated lifetimes of MTs and SQTs were in the range of 6–4800 min and 5–3800 min, respectively (calculated by assuming atmospheric concentration of O$_3$=40 ppb).

These ozone exposure results indicate that ozone removal should be used prior to sampling of terpenes. Several different kind of ozone traps and filters have been used to trap ozone, but some of them are not suitable even for MTs and SQTs (Calogirou et al., 1996; Mermet et al., 2019; Pollmann et al., 2005). In addition, some of the ozone removal techniques have very short ozone removal capacity and they have to be changed frequently and are therefore not ideal for online sampling (Bouvier-Brown et al., 2009; Fick et al., 2001; Mermet et al., 2019). Based on the ozone removal results presented in Hellén et al. (2012) and the inlet recovery results presented in this study (Fig. 2d), heated stainless-steel line could be a good solution for ozone removal especially in online sampling. Alternatively, Mermet et al. (2019) proposed that comparable results to heated stainless-steel line could be obtained with the use copper tubes coated with potassium iodide (KI) for some MTs and SQTs, however, the performance of KI scrubber was not evaluated in regards of DTs.

## 3.2 Real sample application results

As explained in Section 2.4., in the following experiments the plant material samples were heated in order to characterize the DTs potentially emitted by the studied tree species and to further test the TD-GC-MS method performance in view of real sample analysis. The plant material samples were heated to facilitate the release of DTs from the plant storage pools into the gas-phase.

### 3.2.1 Headspace extraction of heated pine needles and spruce twigs

From the dynamic headspace sampling of heated pine needles and spruce twigs several diterpenes could be detected, however, these compounds were detected mainly at the highest temperatures tested and eventually in quite low concentrations. The chromatograms of pine needles and spruce twigs samples are shown in Figs. 6 and 7 and the detected peaks are outlined in Table S4.

In total, 11 hydrocarbon diterpenes ($C_{20}H_{32}$), two oxygenated diterpenes ($C_{20}H_{34}O$) and one unknown BVOC ($C_{19-20}H_{28-30}$) were detected in the headspace samples of spruce twigs (Fig. 7 and Table S4). Of these 14 compounds, only five could be tentatively identified based on mass spectra and retention index matches. The identified compounds were rimuene, cembrene, cembrene A, 13-epi-manool oxide and abietatriene. Most of these compounds have been detected previously in spruce essential oil samples (Norin and Winell, 1972;Wajs et al., 2006), with the exception of rimuene which, however, has been observed recently

in ambient air in Amazon (Yee et al., 2018).

In the pine needles samples, only few DTs could be detected, however, several other similar $C_{18-20}H_{26-32}O_{0-2}$ type of BVOCs were observed (Fig. 6 and Table S4). Overall, three hydrocarbon diterpenes, two oxygenated diterpenes and eight other similar high molecular weight compounds (with potential formulas of e.g. $C_{19}H_{30}$, $C_{18}H_{26}O$ and $C_{20}H_{28}O$) were detected. Of the 13 compounds observed, only three could be tentatively identified. These identified compounds were sandaracopimaradiene, 13-

epi-manool oxide and abietadiene. Of these DTs, only abietadine has been found also previously in pine essential oil samples (Judzentiene and Kupcinskiene, 2008;Judžentienė et al., 2006;Kupcinskiene et al., 2008;Tumen et al., 2010). Most of those previous studies have detected also manool oxide, which is a stereoisomer of 13-epi-manool oxide that we observed here. Interestingly, sandaracopimaradiene has been observed also in the ambient air in Amazon (Yee et al., 2018).

Unfortunately, some of the peaks that had the same (or *almost* the same) retention index values in spruce and pine samples

(Table S4), could not be assigned to be the same compound based on mass spectra. Therefore, it is possible that there might be overlapping compounds in the chromatograms. However, since the mass spectra were different, it is equally possible that the overlapping compounds do not affect the quantification since they seemed to be species specific.

Emission rate coefficients (Fig. S18-S19) were calculated in order to evaluate specifically the relative amount of DTs emitted in comparison to MTs and SQTs. These results are merely semi-quantitative and the uncertainties might be fairly high (e.g. all

detected diterpenes were quantified based on either ent-kaurene or cembrene, and some MTs had higher concentrations in samples than in standards, so the calibration needed to be extrapolated for obtaining mass concentrations). Only the standard analyte MTs and SQTs found in samples were quantified. The results from spruce twigs at 70 °C are not presented due to moisture related problems in the experiments.

As can be seen in Fig. 8, the emission rates were generally roughly 1 to 3 orders of magnitude lower for DTs when compared

to MTs and SQTs (see also Figs. S18-19 for individual compound plots). In general, the emission rates increased as the temperature increased with all compounds. At 30 °C, MTs and SQTs emission rates from pine needles were on average 464±431 and 50±71 ng $g_{dw}^{-1} h^{-1}$, respectively. At the same conditions, the emission rates of DTs were on average only 1±1 ng $g_{dw}^{-1} h^{-1}$, and of the five DTs detected in total, only three were above LOQ at 30 °C. Similarly, the emission rates at 30 °C from spruce twigs were 1052±825 and 164±97 ng $g_{dw}^{-1} h^{-1}$ for MTs and SQTs, respectively. Of the 12 DTs observed in spruce twigs

emissions, only one was above LOQ at 30 °C (at emission rate of 1 ng $g_{dw}^{-1} h^{-1}$). At 60 °C the emission rates of MTs and SQTs were roughly 2-fold higher than those observed at 30 °C in both pine needles and spruce twigs samples, however, the corresponding increase in the emission rates of DTs were roughly 30- to 85-fold (up to 70±129 ng $g_{dw}^{-1} h^{-1}$). Clearly the increase in DTs emissions was more drastic than the increase in MTs and SQTs emissions as the temperature increased (Fig. 8).

### 3.2.2 Branch enclosure experiments

The branch enclosure experiments of a living tree in a boreal forest were conducted in view of the results obtained in the previous section. Since it was though that the DTs emissions would need drastic conditions (e.g. drought, warm temperatures, direct sunlight, pest stress) to be emitted in detectable amounts, the cuvette was heated to high temperatures to promote the DT emissions in order to swiftly test the method in practice. The cuvette inside temperature was heated from ca. 14 to 61 °C in the course of the day (from morning to afternoon, Fig. S5). As mentioned in Sect. 2.4, the sorbent tube sampling was

performed either from a bypass flow or directly from the cuvette outlet port on two different sampling dates. These experiments were conducted in a proof-of-concept type of approach. TD-GC-MS2 was used for analysis since the TD-GC-MS1 was not available at the time of these experiments.

A full list of compounds detected in the spruce emissions are shown in Table S5 and an example chromatogram from a sample analysis in Fig. S20. In total, 25 MTs, 14 SQTs, six DTs and 10 other BVOCs were detected (Table S5). The DTs were mainly

detected once the enclosure temperature was higher than 40 °C and most DTs were detected only at ca. 60 °C. Most of the here tentatively identified compounds have been observed previously in essential oil samples, headspace samples or branch enclosure emissions from spruce (Hakola et al., 2017;Kännaste et al., 2013;Radulescu et al., 2011;Wajs et al., 2006). Similar to the results of spruce twigs in Sect. 3.2.1, the DTs rimuene and cembrene were identified in these branch enclosure experiments. However, the other DTs and alike compounds could not be verified to be matching between the different spruce

sample types. This lack of similarity might be simply due to chemotypic variability between the trees (Bäck et al., 2012;Hakola et al., 2017). The one tentatively identified DT was assigned to be 13-epi-manool, although it might be as well manool (the difference in RI and mass spectra is not that evident between these two stereoisomers), which has been detected previously in spruce samples (Kanerva et al., 2008;Radulescu et al., 2011).

Similar as in the previous section, DTs emission rates were calculated for the relative comparison to MTs and SQTs values.

These emission rate results are merely semi-quantitative and have high uncertainty (e.g. some MT concentrations were above the calibration range), thus the results should be evaluated with caution. The emission rates were quite different on the two sampling dates tested (Fig. 9). On the first sampling date, the MTs, SQTs and DTs emission rates were on average 4784±5845, 321±247 and 67±87 ng $g_{dw}^{-1}$ h$^{-1}$ at 60±1 °C, respectively. On the latter date, the emission rates at 59±1 °C were on average 8024±9712, 641±308 and 419±197 ng $g_{dw}^{-1}$ h$^{-1}$ for MTs, SQTs and DTs, respectively. The MTs and SQTs emission rates were

roughly 2-fold higher on the latter day, whereas the DT emission rates were then approximately 10 times higher than on the first sampling day. In relative terms, the DT emissions were significantly higher on the latter day, for example, the respective DT/SQT-ratio increased from 9 % to 48 % between the sampling dates. It is likely that the difference observed in results between the two sampling dates is due to the stress induced by cutting the branch and the related experimental work on the first sampling date. Nonetheless, in field conditions the difference in DTs emission rates in comparison to MTs and SQTs

emission rates were similar as in the laboratory headspace sample (roughly 2- to 70-fold difference on average at ca. 60 °C).

## 4 Conclusions

An analytical method using sorbent tube sampling followed by TD-GC-MS analysis for the determination of monoterpenes (MTs, $C_{10}$), sesquiterpenes (SQTs, $C_{15}$) and diterpenes (DTs, $C_{20}$) in gas-phase samples was developed. The intermediate precision (3–10 %) and expanded measurement uncertainties (16–55 %) for terpenes were considered to be adequate and fit-for-purpose. The LOQ values for MTs, SQTs and DTs were 13–88 pg, 33–340 pg and 287–518 pg, respectively. If using a sample volume of 5 L in offline sampling, the LOQ values would be approximately 0.5–2.8 pptv for MTs, 0.8–7.5 pptv for SQTs and 5.1–9.3 pptv for DTs.

The sampling recovery results (15 m long Teflon FEP line, 1 m heated stainless-steel line and a Teflon cuvette bag) were mostly acceptable (within 100±20 %) for all studied compounds. For example, the recoveries of DTs were on average 86±1% and 85±6% when the 1 m heated stainless-steel and 15 m Teflon FEP sampling lines were tested, respectively. These sampling recovery results of DTs were promising, since they demonstrated that the DTs were not lost in excessive amounts in the sampling lines tested.

Comparison of online mode sampling and offline mode sampling with TD-GC-MS1 revealed that some of the terpenes are lost in excessive amounts in the online sampling mode. The online mode sampling efficiency could be improved by changing stainless-steel surface to glass in the sample path and by increasing the sample air relative humidity. However, based on the tentative experiments, the online mode TD-GC-MS sampling is only valid quantitatively for monoterpenes and some sesquiterpenes. Diterpenes and caryophyllene oxide results were poor in online sampling mode even with the best possible set up tested (recoveries typically <70 %), limiting the applicability only to semi-quantitative or qualitative analysis. These online mode recovery results highlight the importance of testing each experimental setup in practise prior to conducting campaign measurements.

Based on the ozone reactivity experiments conducted, the literature reported predicted ozone reaction rate coefficients ($k_{O3}$) of some diterpenes might be underestimated. Our results indicated, that the experimentally derived $k_{O3}$ value of ent-kaurene was approximately 2 orders of magnitude higher than the literature predicted $k_{O3}$ value. Consequently, the role of diterpenes in atmospheric oxidation reactions could be more profound than predicted. In addition, the results also highlight the difficulties related to the determination of DTs in ambient air. Theoretically, the emissions of DTs are very low to begin with and the high reactivity decreases the concentrations even further, making the time frame of DT analysis extremely short.

Based on the real sample applications results, we could characterise the DTs emitted by Norway spruce and Scots pine. Multiple DTs and similar high molecular weight BVOCs ($C_{18–20}H_{26–32}O_{0–2}$) could be detected in the emissions, however, only few could be tentatively identified. Of the DTs tentatively identified in this study (e.g. rimuene, cembrene, sandaracopimaradiene, 13-epi-manool oxide and abietadiene), many have been detected previously in essential oil samples and/or even in atmospheric air elsewhere. The DTs emitted by different tree species were partly species specific (e.g. cembrene and rimuene were observed in spruce emissions, but not in pine emissions). It would be beneficial to perform similar experiments with other tree species for comparability, and in order to identify the possible common/specific DT traces.

In general, the semi-quantitatively estimated DTs emission rates were roughly 1 to 3 orders of magnitude lower than the MTs and SQTs emission rates. Diterpenes were mostly emitted at high temperatures (typically ≥35 °C) and the temperature dependence in emissions of DTs was relatively more profound than in those of MTs and SQTs. In principle, the results indicated that high temperatures are needed for instrumentally detectable levels of DTs to be emitted in branch enclosure measurements. These conditions could be met, for example, during direct sunlight exposure and in sunny warm days. In addition, the highest temperatures tested in this study (up to 60 °C) might be reached under extreme situations during atmospheric measurements, such as in the surroundings of forest fires. Long-term measurements in real-life field conditions would be required to truly study the emissions of DTs (in addition to other terpenes and BVOCs) and their atmospheric relevancy.

*Data availability.* Data available upon request by contacting the corresponding author (aku.helin@fmi.fi).

*Author contributions.* AH, HaHa and HeHe designed the experiments, and AH and HeHe carried them out. AH and HeHe performed the data analysis. AH prepared the manuscript with contributions from all co-authors.

*Competing interests.* The authors declare that they have no conflict of interest.

*Acknowledgements.* This work was supported by the Academy of Finland projects (no. 316151 and no. 275608) and Academy of Finland via the Center of Excellence in Atmospheric Sciences (grant no. 307331). The staff of the SMEAR II station are thanked for their collaboration. Toni Tykkä and Anne-Mari Mäkelä are thanked for help in the laboratory.

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

**Table 1. List of analytes, corresponding retention indices (RI) and analytical figures of merit.**

| Compound | CAS Number | Formula | RI[a] | Quan-tify m/z ion | LOQ in pg (in pptv)[b] | Repea-tability (RSD %) | Inter-mediate precision (RSD%) | U[c] (%) | U[d] (%) |
|---|---|---|---|---|---|---|---|---|---|
| α-Pinene (MT) | 7785-70-8 | $C_{10}H_{16}$ | 945 (942) | 93 | 34 (1.2) | 2.6 | 3.8 | 17 | 18 |
| camphene (MT) | 79-92-5 | $C_{10}H_{16}$ | 963 (960) | 93 | 14 (0.5) | 1.6 | 4.0 | 17 | 19 |
| β-pinene (MT) | 19902-08-0 | $C_{10}H_{16}$ | 989 (988) | 93 | 13 (0.5) | 4.2 | 3.2 | 17 | 20 |
| 3-carene (MT) | 498-15-7 | $C_{10}H_{16}$ | 1018 (1017) | 93 | 18 (0.6) | 2.0 | 3.3 | 16 | 18 |
| p-cymene (MT) | 99-87-6 | $C_{10}H_{14}$ | 1033 (1031) | 119 | 20 (0.7) | 2.0 | 4.2 | 18 | 20 |
| limonene (MT) | 5989-54-8 | $C_{10}H_{16}$ | 1039 (1037) | 68 | 62 (2.2) | 1.5 | 3.8 | 17 | 18 |
| 1,8-cineol (MT) | 470-82-6 | $C_{10}H_{18}O$ | 1044 (1042) | 154 | 27 (0.9) | 2.6 | 4.7 | 18 | 21 |
| terpinolene (MT) | 586-62-9 | $C_{10}H_{16}$ | 1094 (1093) | 121 | 65 (2.3) | 2.5 | 5.0 | 19 | 20 |
| linalool (MT) | 78-70-6 | $C_{10}H_{18}O$ | 1100 (1100) | 71 | 88 (2.8) | 2.4 | 4.8 | 20 | 20 |
| 4-acetyl-1-methylcyclohexene (4-AMCH) (BVOC) | 6090-09-1 | $C_9H_{14}O$ | 1140 (1139) | 95 | 120 (4.2) | 2.7 | 6.3 | 21 | 22 |
| nopinone (BVOC) | 38651-65-9 | $C_9H_{14}O$ | 1154 (1153) | 83 | 29 (1.0) | 2.7 | 5.3 | 19 | 22 |
| bornylacetate (BVOC) | 5655-61-8 | $C_{12}H_{20}O_2$ | 1295 (1294) | 95 | 49 (1.2) | 2.4 | 6.2 | 21 | 24 |
| longicyclene (SQT) | 1137-12-8 | $C_{15}H_{24}$ | 1401 (1401) | 94 | 35 (0.8) | 2.4 | 4.9 | 19 | 22 |
| isolongifolene (SQT) | 1135-66-6 | $C_{15}H_{24}$ | 1422 (1422) | 161 | 38 (0.9) | 2.5 | 5.7 | 20 | 23 |
| β-caryophyllene (SQT) | 87-44-5 | $C_{15}H_{24}$ | 1445 (1444) | 93 | 105 (2.5) | 2.2 | 3.5 | 18 | 23 |
| β-farnesene (SQT) | 18794-84-8 | $C_{15}H_{24}$ | 1457 (1456) | 69 | 198 (4.7) | 5.0 | 5.9 | 23 | 24 |
| α-humulene (SQT) | 6753-98-6 | $C_{15}H_{24}$ | 1481 (1480) | 93 | 33 (0.8) | 2.1 | 4.2 | 19 | 25 |
| caryophyllene oxide (SQT) | 1139-30-6 | $C_{15}H_{24}O$ | 1612 (1612) | 79 | 340 (7.5) | 18.9 | 8.6 | 26 | 30 |
| cembrene (DT) | 1898-13-1 | $C_{20}H_{32}$ | 1959 (1959) | 93 | 287 (5.1) | 4.0 | 9.5 | 32 | 55 |
| ent-kaurene | 562-28-7 | $C_{20}H_{32}$ | 2106 | 257 | 518 | 2.9 | 7.7 | 27 | 42 |

| | | | | | | | | |
|---|---|---|---|---|---|---|---|---|
| (DT) | | | (2106) | | (9.3) | | | |
| 3-methylene-5-α-androstane (3-MA)[e] (DT) | not assigned | $C_{20}H_{32}$ | 2153 (2152) | 257 | 383 (6.9) | 3.0 | 7.2 | - - |

[a] RI as Kovats indices (and arithmetic indices in brackets)

[b] LOQ in pptv calculated by using sample volume of 5 L (at T=298 K and p=1 atm)

[c] U (%) for ambient air application

[d] U (%) for branch enclosure emissions application

[e] The purity of this crude standard was set to being 100% (allowing the determination of LOQ, but unknown uncertainty).

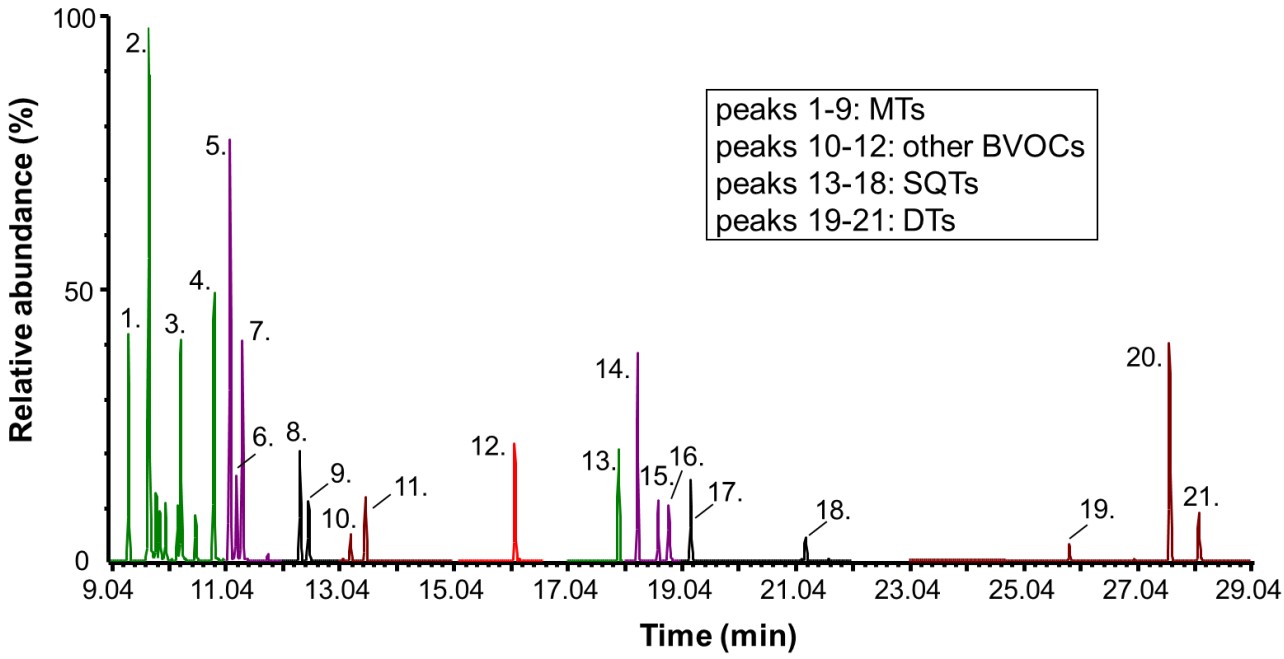

**Figure 1. Overlaid selected ion recording (SIR) chromatograms of a standard solution (c=10-50 ng in sorbent tube) analysed by TD-GC-MS1. The analysis conditions are presented in Table S2. Peak identification can be made based on Table 1, from low RI (peak 1. α-pinene) to high RI (peak 21. 3-MA).**

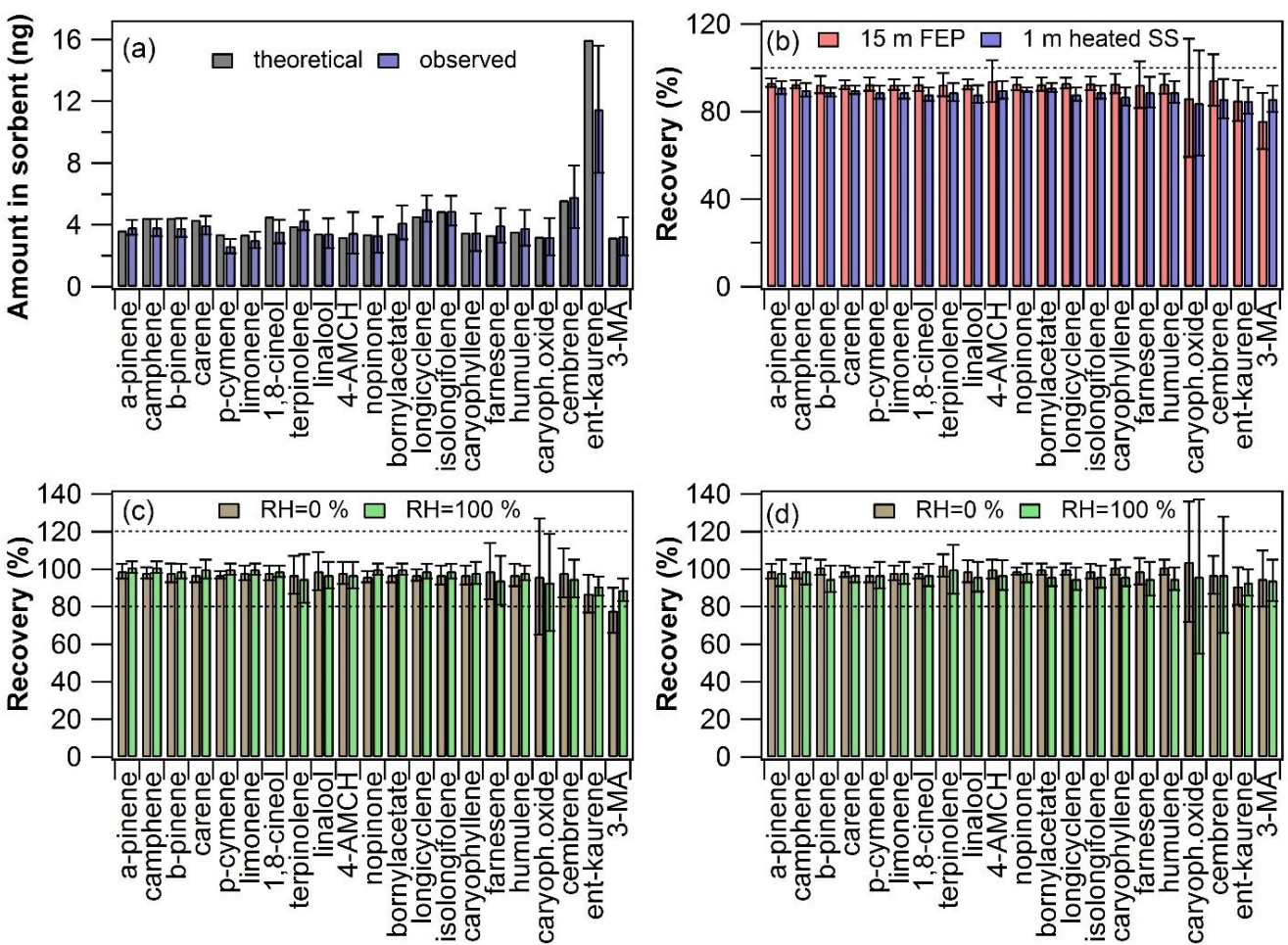

Figure 2. Inlet recovery experiment results: (a) calculated theoretical amount and observed amount in the first sorbent tube (*n*=18), (b) inlet recovery levels at laboratory room conditions (*n*=9 with both inlets), (c) inlet recovery levels at different RH levels with Teflon FEP inlet (*n*=8-10) and (d) with heated stainless-steel inlet (*n*=9-10). The error bars represent reproducibility (*n*=8-18).

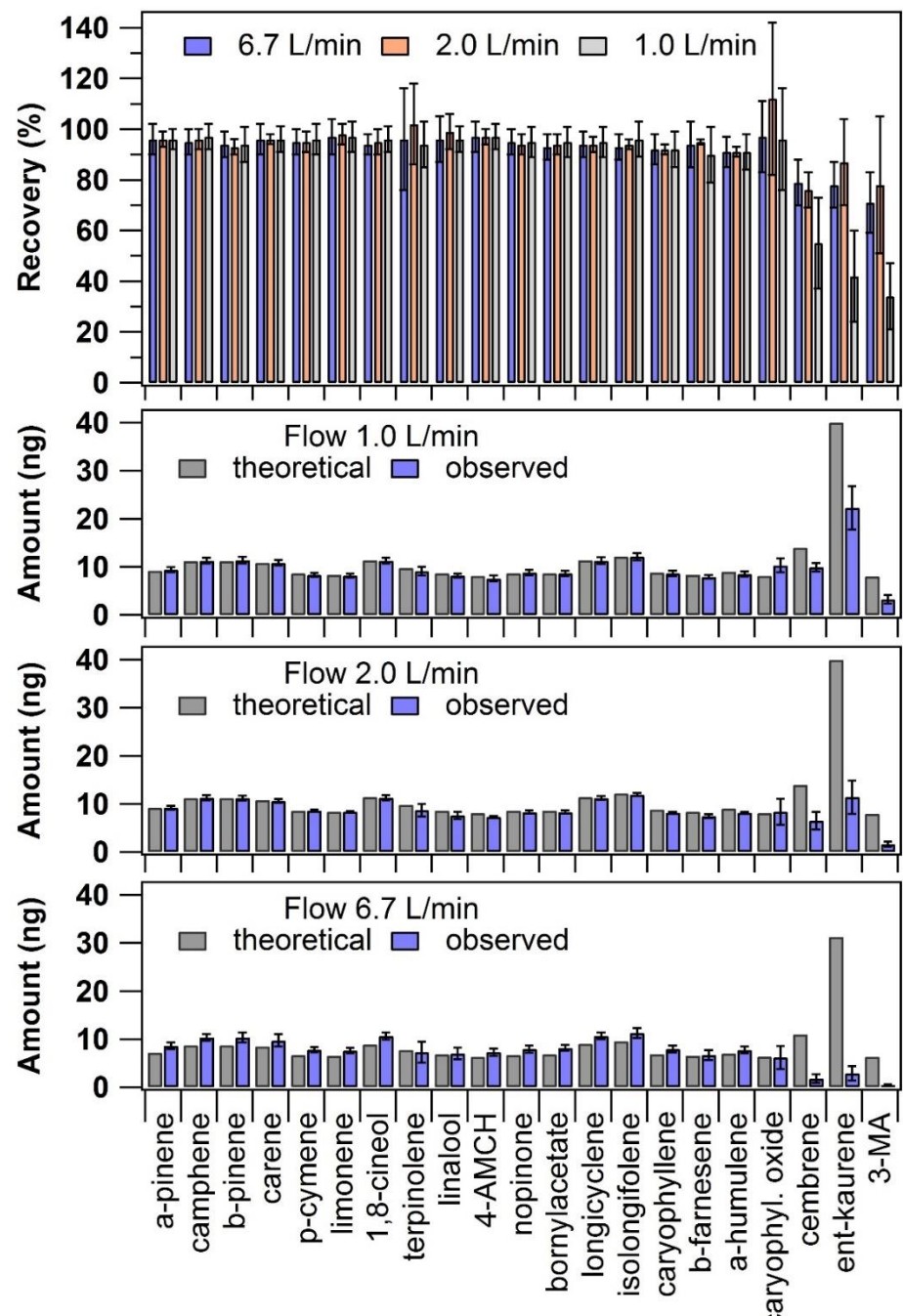

Figure 3. Cuvette recovery results (top panel plot) and the absolute amount detected in the first sampling sorbent tube presented according to the different inlet flow rates used (the lowest three panel plots). The theoretical expected amount was calculated based on the used experimental conditions. The error bars represent reproducibility ($n$=3-9).


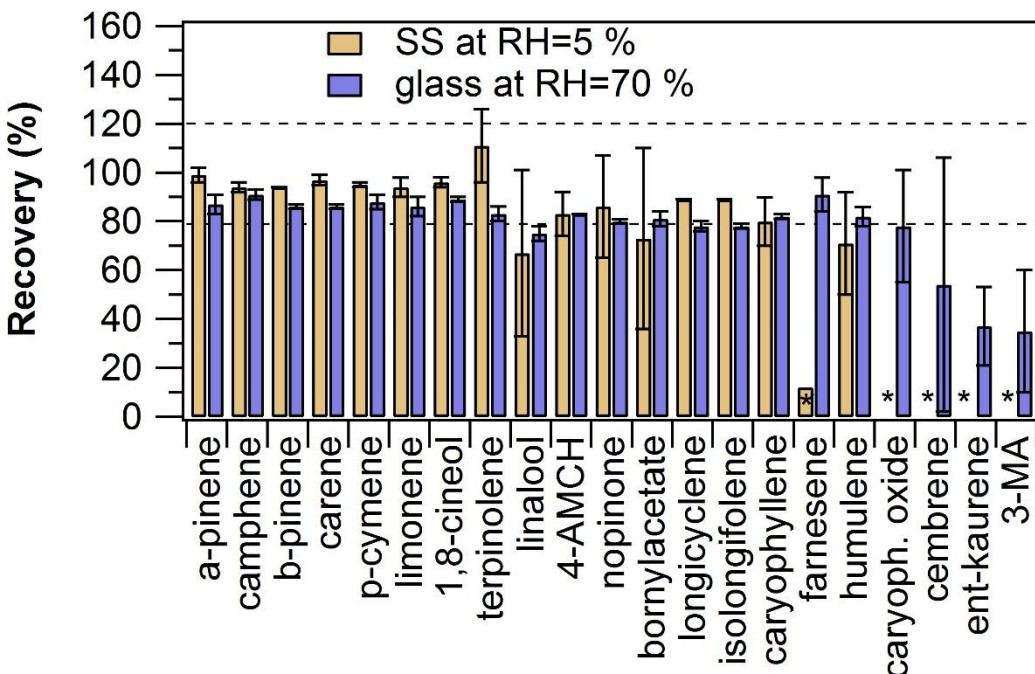

**Figure 4.** Comparison of online sampling mode recoveries obtained by using either empty stainless-steel (SS) tube (at RH=5 %) or empty glass tube (at RH=70 %) in online sampling. The empty spaces marked with asterisk indicate <LOQ results in online sampling. The error bars represent repeatability (*n*=2).

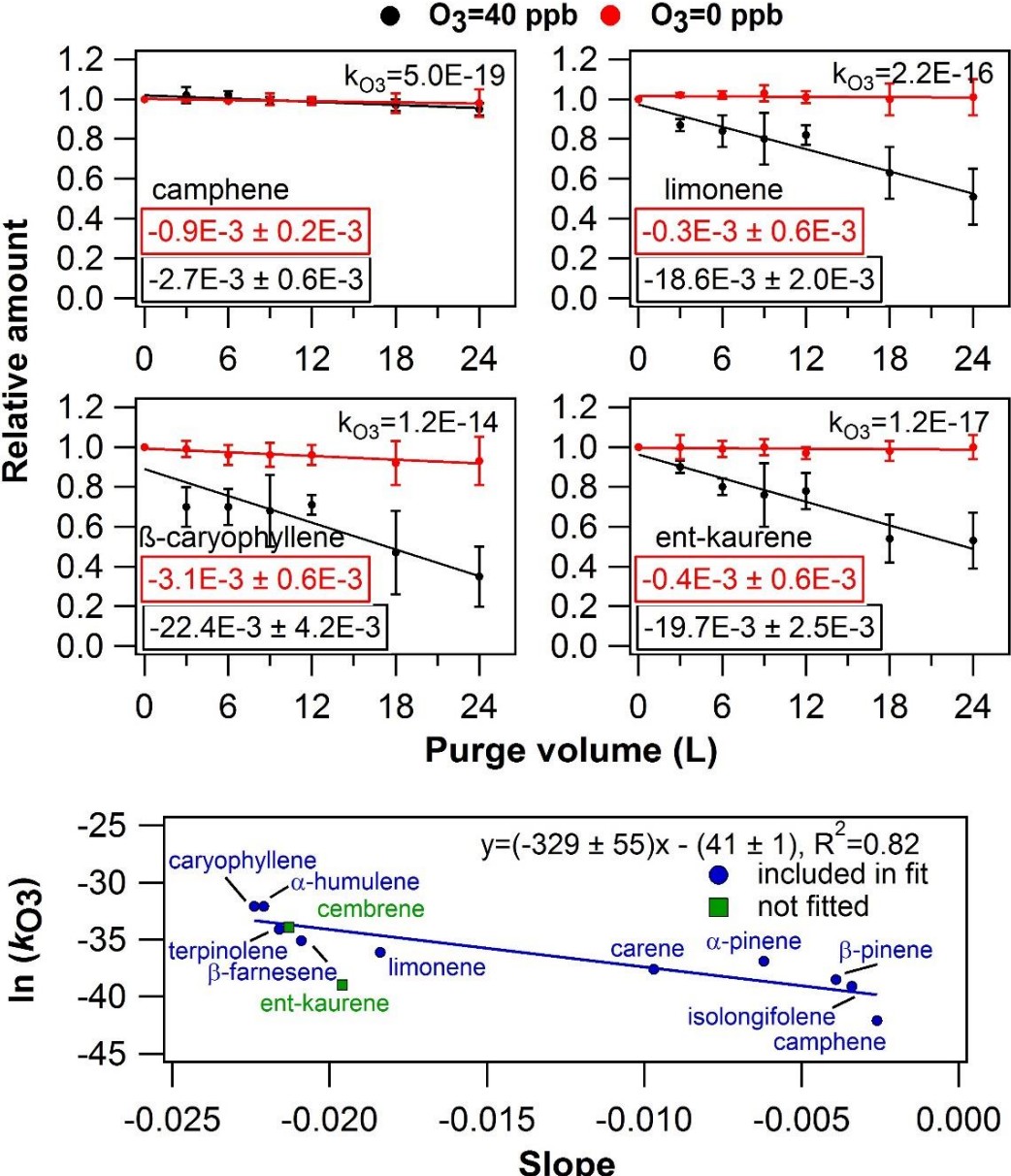

**Figure 5. Results from ozone exposure experiments. In the upper panel plots are selected examples from the ozone exposure experiments presented as relative amount of compound in sorbent tube as a function of purge volume. The values embedded into the plots are the linear regression fit slopes (±uncertainty) of corresponding conditions (either $O_3=0$ ppb or $O_3=40$ ppb). The compound specific $k_{O3}$ values are presented in the plots as well. The error bars represent reproducibility ($n=3$). In the lower panel plot is presented the literature retrieved $\ln(k_{O3})$ values (see Table S3) as a function of the linear regression fit slopes.**

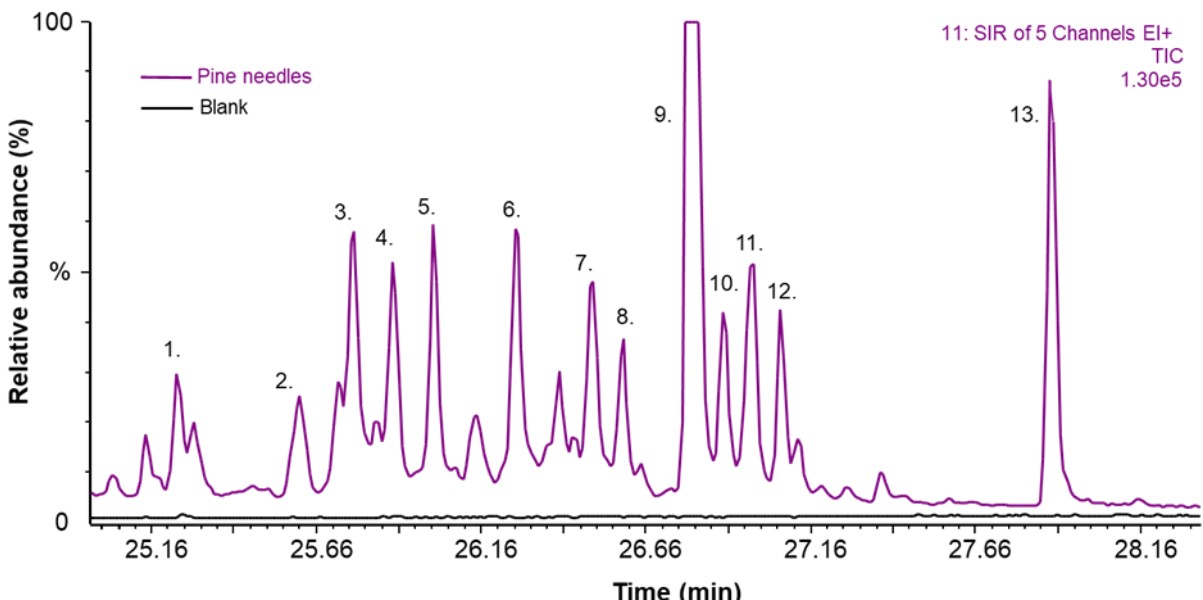

**Figure 6. Selected ion recording (SIR) chromatograms from the TD-GC-MS1 analysis of pine needles and blank dynamic headspace sorbent tube samples (sampling at 60 °C for 20 min at a flow rate of 100 mL/min). The five m/z ions included in the SIR were m/z 93, 229, 257, 272 and 275. The chromatogram is scaled for presentation purposes, peak number 9 is not showed in full height in order to visualize the other peaks more clearly. Peak identification: 1. Unknown ($C_{19}H_{30}$, BVOC), 2. Unknown ($C_{19}H_{28}$, BVOC), 3. Unknown ($C_{18}H_{26}O$, BVOC), 4. Unknown ($C_{19-20}H_{32}O_{0-2}$, BVOC), 5. Unknown ($C_{20}H_{32}$, DT), 6. Sandaracopimaradiene ($C_{20}H_{32}$, DT), 7. Unknown ($C_{19-20}H_{28}O_{0-1}$, BVOC), 8. Unknown ($C_{20}H_{30-32}O_{0-2}$, BVOC), 9. 13-epi-manool oxide ($C_{20}H_{34}O$, DT), 10. Unknown ($C_{19-20}H_{28-32}O_{0-1}$, BVOC), 11. Unknown ($C_{19-20}H_{28}O_{0-1}$, BVOC), 12. Unknown ($C_{20}H_{34}O$, DT) and 13. Abietadiene ($C_{20}H_{32}$, DT). See Table S4 for further information on peak identification.**

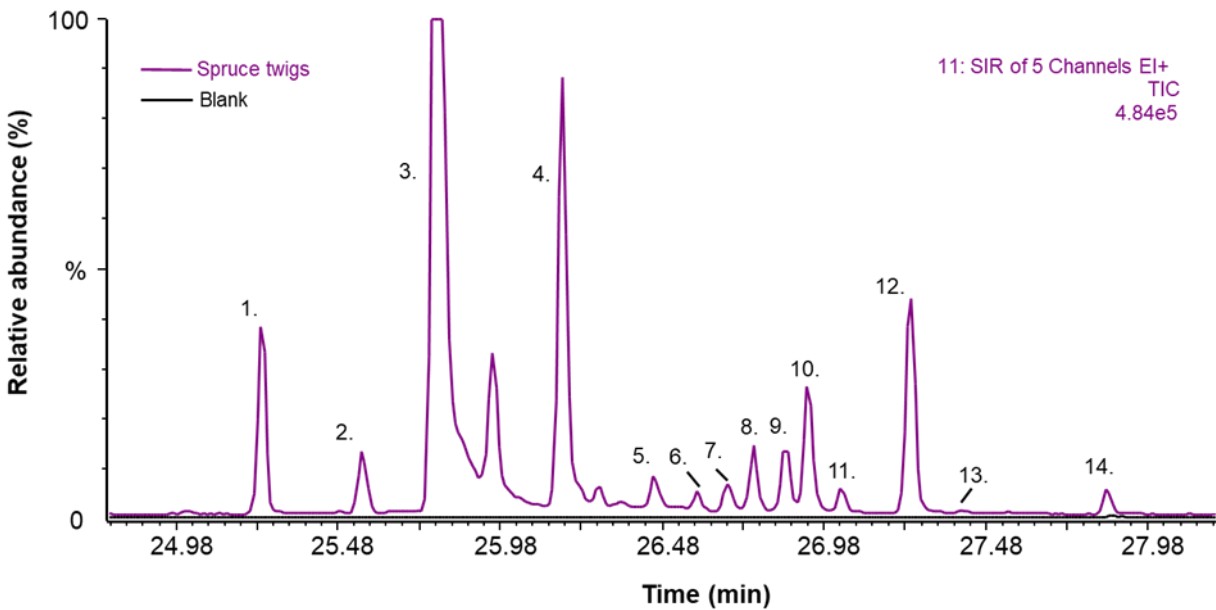

**Figure 7. Selected ion recording (SIR) chromatograms from the TD-GC-MS1 analysis of spruce twigs and blank dynamic headspace sorbent tube samples (sampling at 60 °C for 20 min at flow rate of 100 mL/min). The five m/z ions included in the SIR were m/z 93, 229, 257, 272 and 275. The TIC is scaled for presentation purposes, peak number 3 is not showed in full height in order to visualize the other peaks more clearly. Peak identification: 1. Rimuene ($C_{20}H_{32}$, DT), 2. Unknown ($C_{20}H_{32}$, DT), 3. Cembrene ($C_{20}H_{32}$, DT), 4.**

**Cembrene A ($C_{20}H_{32}$, DT), 5. Unknown ($C_{20}H_{32}$, DT), 6. Unknown ($C_{20}H_{32}$, DT), 7. Unknown ($C_{20}H_{32}$, DT), 8. 13-epi-manool oxide ($C_{20}H_{34}O$, DT), 9. Unknown ($C_{20}H_{32}$, DT), 10. Unknown ($C_{19-20}H_{28-30}$, BVOC), 11. Unknown ($C_{20}H_{34}O$, DT), 12. Unknown ($C_{20}H_{32}$, DT), 13. Abietatriene ($C_{20}H_{30}$, DT) and 14. Unknown ($C_{20}H_{32}$, DT). See Table S4 for further information on peak identification.**

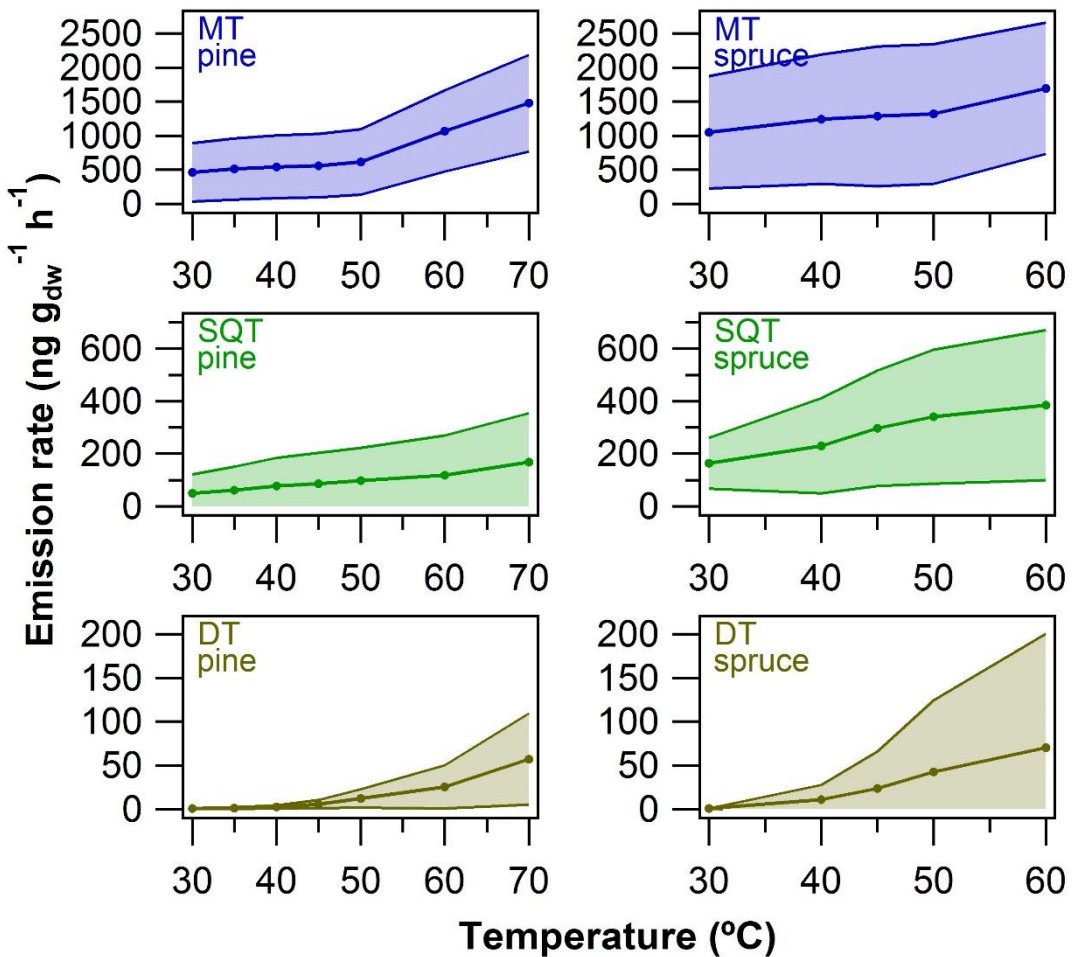

**Figure 8. Terpenoid semi-quantitative emission rates from pine needles and spruce twigs as a function of oven temperature in the dynamic headspace extraction experiments. The average (± standard deviation) emission rates MTs, SQTs and DTs were calculated based on data presented in Figs. S21-S22.**

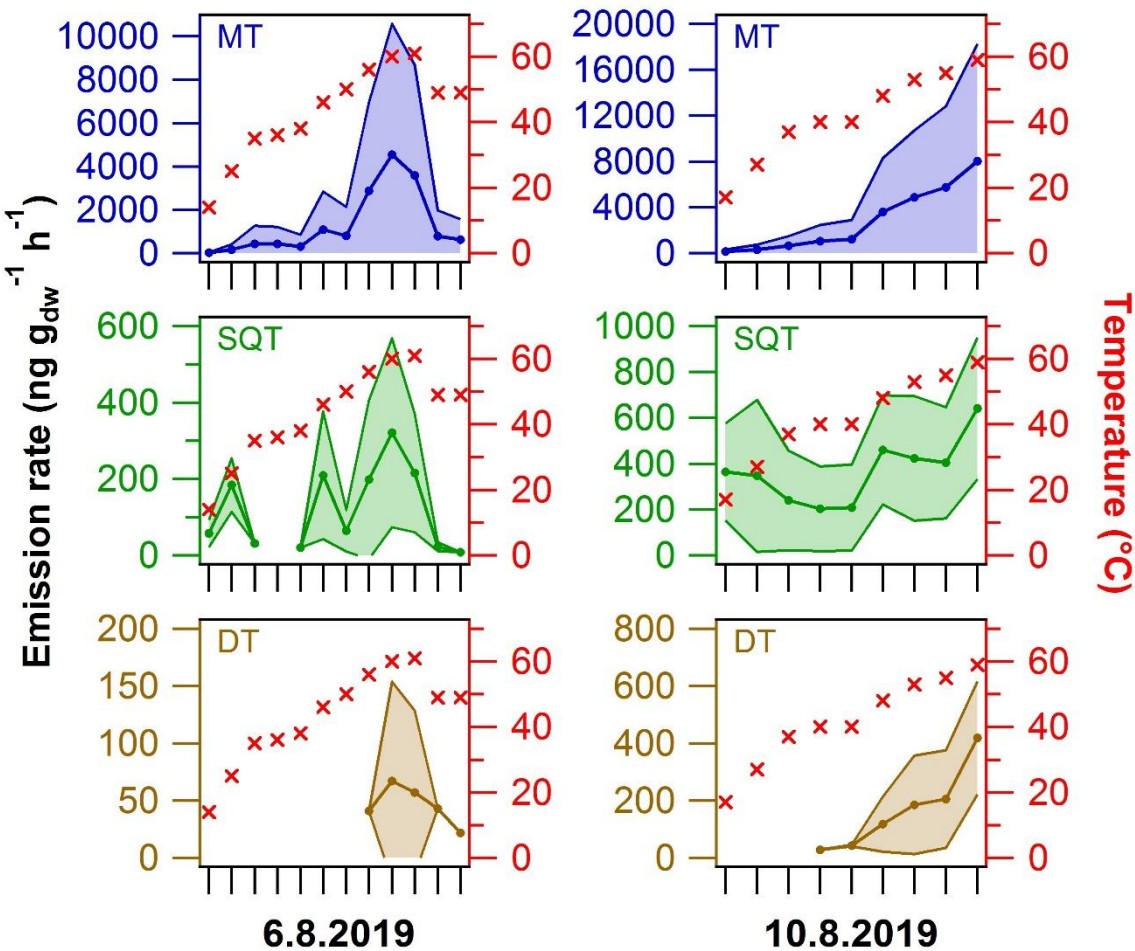

**Figure 9. Semi-quantitative average (±standard deviation) emission rates of MTs, SQTs and DTs from the living spruce branch enclosure experiments on two different sampling dates. The cuvette inside temperature is shown on the right y-axis (for details see Fig. S5). For details on the terpenes included, see Table S5.**