# Peer review of "Optimisation of a thermal desorption-gas chromatography-mass spectrometry method for the analysis of monoterpenes, sesquiterpenes and diterpenes"

_Atmospheric Measurement Techniques, 2019_

## Referee Comment (RC1) · Anonymous Referee #1 · 27 Feb 2020

The manuscript of the development of a thermal desorption gas chromatography mass spectrometry method for the analysis of monoterpenoids, sesquiterpenoids and diterpenoids is a useful contribution to AMT, but it needs to undergo major revisions before it is published.

There are three main issues 1- The method has already been validated for monoterpenoids and sesquiterpenoids, so all the information in the paper about these compounds is not necessary. Furthermore, there is a strong mix up in the text between monoterpenoids, terepenes and terpenoids which is highly confusing. Since I recom-

mend to use only diterpenoids, this problem will be solved.

2- The ambient measurements are not representative. You will never get 60°C in a boreal environment. Additionally, there is no blank for these measurements, making the experiments quite doubtful. Please discuss why such high increases of temperature are representative for boreal environments, and how likely is for such compounds to really be emitted from this ecosystem.

3- Method development. The paper needs definitions on method development and validation. Please explain how this is different from previous studies. Furthermore, please discuss while you develop a method for quantification but at the end, for real samples you use a different system and you only give semi-quantitative information.

Specific comments: Page 1 Line 14: Despite I think ozone reactivity of diterpenes is important and should be stated in the paper, it is really not introduced in the text. . . how these measurements add to your method validations. It must be better introduced here in abstract and in text.

Page 1 line 19: I do not understand why heating up to 60°C is representative of anything natural, you will never get these temperatures in boreal ecosystems. In which way these diterpenes will then be emitted to ambient air?

Page 1 Line 20: five what?

Page 1 Line 20: it is really not acceptable that you provide a method validation to then deliver semi-quantitatively.

Page 2 Line 41. The way you express monoterpenoids (as well as sesquiterpenoids and diterpenois) is really confusing. By omitting mt and sqt this problem will be solved. But please do explain what do you mean by monoterpene and monoterpenoid. . . for instance, is p-cymene not a monoterpenoid? And it is only C10H14. . . .In fact, you could name them in Table 1, S4,S5.

Page 2 line 62. This is confusing, please state which branch enclosure you are referring

and which ambient studies? Do you mean you experiments or the previously reported studies?

Page 2 Line 63. If you are going to say this, you need to explain what do you mean by method validation. Do you need two different analytical systems to prove your concentrations? And what do you mean by performance parameters? GC and sampling settings? In is normally well mentioned in the methods of each study? Please state why you method is 1) better than previously mentioned for DT measurements, and 2) how you truly validate the method rather than say your method is good because I do not loose DT in the lines. . .

Page 3 Line 69-78. In your objectives are somehow confusing. First you talk about the applicability of sorbent tubes and GCMS for mt, sqt and dt. I do not know what is new here. This has been shown in the past, plus this is no objective, but rather telling what you did. Later you say the method was incorporated from previous studies for sqt and mt. Thus there is no novelty on sqt and mt, so this builds my point of removing mts and sqt from the text. Then you say you do a method development for dt, and it is not clear to me what is the actual development (not just simply that you have calculated the LOD. . .. But rather than the lod is lower than using this other method. . ..) as compared to mt and sqt analysis. It is also very strange to me that you offer a validated method and then you report only semi-quantitative values. Thus the whole objectives section must be rewritten. Please bear in mind to put what do you intent to do and why.

Page 3 Line 83: If one of the dt standard is not analytically valid, then you have to skip it.

Page 3 Line 84-86. Delete sentence

Page 3 Line 88: Please state why do you dilute the standards in methanol and not other solvent such as hexane.

Page 3 line 88 and 90. You can't report such a large range in concentrations for analyte

solutions. You must be certain of the concentration of your solution, and this is why you use calibrated pipettes. So please change accordingly. If you have a range of concentrations due to different compounds, then state them in a table.

Page 3 line 91. Why do you store solutions in the dark at 4°C?

Page 4 line 103-104. I do not understand this sentence. Please rephrase thinking what is the point of this statement to the objective of the paper.

Page 5 Line 130: why you changed the method (different final state) for TDGCMS2?

Page 5 Line 136-140: I think this information is useless, either rephrase or remove.

Page 5 Line 146: please state how many tubes were stored.

Page 6 line 186: please state which are the guidelines so the reader doesn't need to go somewhere else to find it. You can do it in SI if you don't want this info in the text.

Page 7 line 206: at what temperature did you store it? Then you freeze the material? is not clear how you did it.

Page 7 line 214: what do you mean as a proof-of-concept type of experiment? And please state if the branch enclosure was just a branch or it also had needles in.

Page 7 line 215: What do you mean but shrank?

Page 7 line 217: state reason for copper tube.

Page 7 line 219: what is the reason of artificially promote BVOC emissions? Why would you do this if it is not realistic to the ecosystem of interest.

Page 7 application to real samples. I do not see any blank measurements, please include them. If you have not taken blank measurements then I ask how do you exclude contamination effects.

Page 8 line 231: what is the sufficient confidence?

Page 8 line 233-5: if you can't identify the compounds then remove them from identified compounds.

Page 8 line 241: What is C1?

Page 8 line 348-9. Remove this sentence, it does not contribute anything to the ms.

Page 9 line 250: what is the desorption efficiency of TDGCMS2?

Page 9 line 251: what do you mean by carryover?

Page 9 line 269: why did you choose 60 and not 70 or 50C? How did you know all liquid goes into the gas phase?

Page 9 line 271: how did you observe that diterpenes were sticking into walls and not decomposing some other way?

Page 9 line 273-279: This is a very serious statement which I think is wrong. It is fine that you did not find any losses with you inlet line, but this will be highly dependent upon the ambient temperature that the line will be subjected to. For instance 15 m non heated Teflon line for outdoor sqt monitoring in a boreal ecosystem during winter HAS to be heated!. Please remove statement about not heating the Teflon tube, it is totally misleading.

Page 9-10 lines 280-297: I wonder what will be the effect of transpiration inside a cuvette. This must be really important for diterpenes as you mention, so I think it is wise to let people know that inside cuvette humidity monitoring is essential.

Page 10 line 288-290: why do you think there is a longer residence time in the cuvette for diterpenes? I think this whole paragraph needs to be rewritten. You simply say that you may not be volatilizing well your diterpenes, thus, you can't be really validating the method if you are not sure you can't get all your diterpenes into the gas phase. Please rephrase by 1) taking away the mt and sqt info (because as you say it is explained in other studies) 2) you think account for this HUGE limitation and how this affects your

method validation. In fact, somewhere in the text you must state why you are truly validating a method here with these limitations.

Page 10 line 299. I do not understand how you say first that you can't really be sure of 3MA and then you actually show LOQ values. How is this possible?. Please explain.

Page 10 line 300: please state that RSD means relative standard error. I wonder why you did not show any repeatability among different systems. This will really validate your method!

Page 10 line 306: What do you mean by assuming sampling volume? You must know this.

Page 11 line 327-9: I do not understand this sentence. Please rephrase.

Page 12 line 375. Please expand on why heated stainless steel better than other ozone removal mechanisms.

Page 13 line 380. I think it would be wise and enriching the ms if you add this data.

Page 13 section 3.2.1. Please explain why you heat needles and twigs.

Page 13 line 389. If you could only identify 5 how can you report 11?

Page 13 line 406: what do you mean by coefficients? Please explain.

Page 15 line 460: what do you mean by good? This is very vague, good as compared to what. Please expand.

Editting comments Page 2 Line 34. OH must go before hydroxyl not before radical.

Page 2 line 35: You must explain what VOC is, you only said BVOC before.

Page 7 line 219: put a before temperature and I suppose you mean that you place the temp sensor in the cuvette not the logger itself, right?

Page 9 line 268: replace "with" with "for". And state which compounds you refer to.

---

## Referee Comment (RC2) · Anonymous Referee #2 · 4 Mar 2020

The study by Helin et al. describes in depth the analytical method of a thermal desorption-gas chromatography-mass spectrometry (TD-GC-MS) to quantify a series of Biogenic Volatile Organic Compounds (BVOC). While the core analytical method has been previously used extensively, such a detailed description of the processes and parameters affecting the sampling collection is a valuable addition. The authors focus their investigations to diterpenes, a highly understudied class of compounds that exhibits difficulties in quantification due to their volatility, low abundance, sticky nature and experimental artefacts. The paper reads well but few critical points deserve more

attention. Therefore, the paper is suitable for AMT and I could recommend publication but only after addressing the following comments/concerns.

General comments

1. There seems to be a confusion on the terminology that starts from the title itself and extends to various points of the text. Terpenes are a class of compounds that include monoterpenes, sesquiterpenes and diterpenes. However, it's true that in the literature terpenes and terpenoids are frequently used in the same bases. But monoterpenoids, sesquiterpenoids and diterpenoids are actually modified terpenes, with different chemical formulas (eg. Chan et al., 2016). I would recommend changing the title to the more general "Biogenic Volatile Organic Compounds" or "Terpenoids". Consequently, individual corrections on the text over case-based referral to the individual species is needed.

2. My most serious concern has to do with the striking claim that there is no need for heating the Teflon lines in field applications, in contrast to almost all experimental approaches in the field to date. That conclusion was derived performing sampling recovery tests over which a second sample was taken subsequent to the original (L159: "Inlet line recovery was calculated simply by the relationship of analytes found in the second tube divided by the amounts in the first tube.") One would naturally expect that when the line is kept at the same temperature, artifacts cannot be evaluated properly. The losses on the line could be better illustrated if after the first sample, a subsequent "blank" sample over a now heated line is taken. In this approach, the amount of VOCs stuck in the tube could be quantified and the proper temperature for heating up the lines could be ascertained.

3. Similar to the previous comment, the authors describe in S1.5 the problems that arise with the replacement of the Teflon line. It seems that a new line is causing more artefacts comparing to a used one, indicating some saturation over the line surfaces. In general, the inlet line experiments should be explained in greater detail. The claim

that Teflon lines should remain unheated for field applications has the potential to mislead future research and therefore it cannot become acceptable unless strong and well described proof of concept is presented.

4. There is a lot of useful information, results and discussion in the supplementary information. Sometimes, it becomes challenging to follow the text while browsing through Figures, Supplementary Figures, additional discussion and references. I would recommend to bring forward the supplementary discussion, including several of the figures.

5. In order to derive some quantitative emissions of diterpenes from needles, the experiments (both in lab and field) were performed under unrealistically high temperatures. While this is acceptable for validating the method, it should be further noted that these are stored compounds coming for the pools of the plant material and further highlight the reasons for this approach. Some investigations between the vapor pressure and measured emissions would be nice addition to the paper, if possible.

Specific comments

L34-40. The authors probably refer to the comparative reactivity method. If so, the reference of Kovacs and Brune is misplaced. Instead the study by Sinha et al (2008) should be cited.

L93. Please make a comment on how the commercial adsorbent tubes compare with the custom-made ones.

L154-155. Zero air generators and carbon cartridges have frequently shown that are not completely scrubbing the VOCs. Have you tested the efficiency of these methods? How do they compare between them?

L182 & L239. Please use equations in separate lines so they are clearly distinguishable from the rest of the text.

L375. O3 removal. Have you tested other scrubbers? Why the heated stainless steel in the optimal solution of ozone removal and not e.g. KI scrubbers or Na2S2O3 infused

quartz filters? Please refrain from strong claims that are not supported by experiments.

L386. Please make a comment on the saturated peaks of Sup.Fig.19-20.

L463. Similar to my major comments, it has not been sufficiently demonstrated that the ambient sampling DTs does not require heating lines. One potential experiment to help towards this direction would be to sample through a long inlet line the analyte mixture at different temperatures and present a plot of inlet temperature vs signal.

References

Chan, A. W. H., Kreisberg, N. M., Hohaus, T., Campuzano-Jost, P., Zhao, Y., Day, D. A., Kaser, L., Karl, T., Hansel, A., Teng, A. P., Ruehl, C. R., Sueper, D. T., Jayne, J. T., Worsnop, D. R., Jimenez, J. L., Hering, S. V., and Goldstein, A. H.: Speciated measurements of semivolatile and intermediate volatility organic compounds (S/IVOCs) in a pine forest during BEACHON-RoMBAS 2011, Atmos. Chem. Phys., 16, 1187–1205, https://doi.org/10.5194/acp-16-1187-2016, 2016.

Sinha, V., Williams, J., Crowley, J. N., and Lelieveld, J.: The Comparative Reactivity Method – a new tool to measure total OH Reactivity in ambient air, Atmos. Chem. Phys., 8, 2213–2227, https://doi.org/10.5194/acp-8-2213-2008, 2008.

---

## Author Comment (AC1) · 3 Apr 2020

We thank the reviewer for the evaluation of the manuscript and for raising the awareness of the shortcomings in it. The manuscript was revised according to the reviewers' comments. Here the point-by-point answers are given. The reviewers' comments are in bold and our answers are in normal font. The answers are divided into author's response (AR) and author's changes to manuscript (ACM).

**The manuscript of the development of a thermal desorption gas chromatography mass spectrometry method for the analysis of monoterpenoids, sesquiterpenoids and diterpenoids is a useful contribution to AMT, but it needs to undergo major revisions before it is published.**

**There are three main issues:**
**1- The method has already been validated for monoterpenoids and sesquiterpenoids, so all the information in the paper about these compounds is not necessary. Furthermore, there is a strong mix up in the text between monoterpenoids, terepenes and terpenoids which is highly confusing. Since I recommend to use only diterpenoids, this problem will be solved.**

AR: These were very useful comments and helped us to improve the manuscript. However, we do not fully agree with the reviewer that the information regarding monoterpenes (MTs) and sesquiterpenes (SQTs) should be excluded from the manuscript. The reasons for this will be explained here and in few points later on. Parts of the reviewers' remarks were maybe caused by the lack of appropriate information we had provided and parts perhaps by the improper terminology we had used occasionally in the original manuscript. These parts have now been revised and improved.

The analytical procedure that we have used in this manuscript has not been validated previously as such for the determination of MTs and SQTs in gas-phase samples. We did apply many parts of the previously developed and optimized in-house analytical procedure for our work. These applied parts of the TD-GC-MS method included, e.g. the thermal desorption units' parameters such as desorption temperature, desorption time, desorption flow rate, cold trap temperature and outlet split flow (although, of course, the desorption efficiency was separately tested in this current work and the desorption time was subsequently increased for TDGCMS2). However, in this work we used different analytical column than previously in gas chromatography (GC), and therefore also different GC oven temperature program. Due to this, also the mass spectrometry (MS) selected ion recording (SIR) time windows were changed from the previously in-house validated TD-GC-MS method. The GC column used previously had the same stationary phase material, but the film thickness was in that 1 μm whereas we used here 0.25 μm. Less thick stationary phase was selected to elute the DTs in reasonable time and to minimize the column bleed since higher GC temperatures were needed to elute DTs. Nonetheless, many practical aspects of the procedure could be applied from previous works, such as calibration standard preparation, calibration working range, sorbent tube conditioning etc. Therefore, in the original manuscript, we wanted to be transparent that the procedure was partly applied from our groups' previous works, but naturally this was understood in the way that it was the same method and just diterpenes were added as analytes. As explained above, this was not the case. Consequently, this has been made clearer in the revised manuscript. Now the reader does not need to anymore browse to the prior published articles to see the differences in methods, as the main changes have been explained in the revised manuscript.

In our opinion, it is necessary to present the results also for MTs and SQTs for comparability purposes. In many sections of manuscript, the results of DTs are compared specifically to the results of MTs and SQTs. Furthermore, we do not fully agree with the reviewer that the information about MTs and SQTs is not necessary, since even a similar method development manuscript with only MTs and SQTs still would have some novelty. For example, just recently in Atmospheric Measurement Techniques journal an article covering the method development for the online GC-MS analysis of MTs and SQTs was published (Mermet et al., 2020). Thus, there is still novelty for presenting detailed results about

the method performance results for MTs and SQTs. In our opinion, one of the strengths of our manuscript is that it presents collectively the results for MTs, SQTs and DTs.

The presentation of terpene classification has been changed (also Reviewer #2 had a comment regarding this). As per Reviewers' later comments, the classification of compounds have been added to the tables and plots in the manuscript. The changes are demonstrated later on here in the answers.

ACM: In the Introduction, the following clarification was done:
"In this study, sorbent tube sampling followed by TD-GC-MS analysis for the determination of MTs, SQTs and especially DTs in gas-phase samples was developed and evaluated. The TD-GC-MS method was partially incorporated from our group's previous studies (see e.g. Hellén et al., 2018), and modified accordingly to fit the needs for the analysis of low-volatility diterpenes. In principle, similar analytical procedures have been used previously in both ambient air and enclosure emission studies for the analysis of relatively volatile and semi-volatile MTs and SQTs in multiple field campaigns (Aaltonen et al., 2011;Hellén et al., 2018;Joensuu et al., 2016;Mäki et al., 2017;Vanhatalo et al., 2018). In this work, we build on our previous knowledge of the TD-GC-MS technique and developed our core TD-GC-MS method further to allow the analysis of the fairly low-volatility DTs. In this study, the method development and applicability related experiments done in view for the analysis of DTs included e.g. desorption efficiency, sampling recovery, stability tests, comparison of online and offline sampling modes and ozone reactivity tests. The method analytical figures of merit were also determined, including the quantification limits, intermediate precisions and measurement uncertainties for the terpenes. In addition, qualitative and semi-quantitative experiments were conducted in laboratory and in field conditions to characterise the DTs potentially emitted by boreal forest tree species."

In Section 2.2., the following was added as first paragraph:
"As mentioned before in the Introduction, the TD-GC-MS method used here was partially applied from a previously developed and validated in-house method that has been used in our group's previous studies for the analysis of MTs and SQTs (see e.g. Hellén et al., 2012, 2018). In this current work, the analytical GC column stationary phase (nonpolar 1,4-bis(dimethylsiloxy)phenylene polydimethylsiloxane) film thickness was decreased from a previously used thick phase 1.0 µm column to 0.25 µm column as a compromise to maintain proper selectivity between the terpenes and to elute the DTs in reasonable time (chromatograms presented later on). It was expected that the DTs might retain too strongly in the 1.0 µm film thickness column and that the column bleed could become a problem in long-term use since higher temperatures are required to elute the DTs. Thus, the analytical column was replaced, GC oven temperature program was modified, and the MS scan settings were updated and modified. For the most part, the TD unit parameters' were kept as in the previously optimized in-house method, however, the desorption efficiency was separately verified in this work."

**2- The ambient measurements are not representative. You will never get 60_C in a boreal environment. Additionally, there is no blank for these measurements, making the experiments quite doubtful. Please discuss why such high increases of temperature are representative for boreal environments, and how likely is for such compounds to really be emitted from this ecosystem.**

AR: In general, we do not state that the high temperatures are representative of natural conditions. High temperatures were tested in order: i) to test the developed method in practice in real sample analysis, ii) to identify the DTs potentially emitted by the studied plant material samples and iii) to quantify the emissions of DTs and to compare those in relative terms to MTs and SQTs emissions. In the revised manuscript, we have elaborated the decision to use heating in the experiments more clearly (see ACM).

Given that there are no direct results yet about the presence of DTs in boreal environments, we cannot say and do not want to speculate how these compounds are being emitted and if they are being emitted in detectable amounts. It is somewhat out of the scope of our paper. However, we have added information that previous studies have concluded that the DTs are likely emitted from the storage pools of the plants and that the emissions have been seen to be temperature driven (Haberstroh et al., 2018;Matsunaga et al., 2012;Yáñez-Serrano et al., 2018). The diterpenes were not solely emitted at 60 °C, e.g. as written in Section 3.2.1 regarding the emissions from pine needles "…and of the five DTs detected in total, only three were above LOQ at 30 °C…". Diterpenes were detected at lower temperatures also, but in less amounts as can be seen in Figs. 8 and 9.

In addition, please note that the developed analytical procedure is not restricted to be used only in boreal environments. As it currently stands in the manuscript, the first line in abstract: "*In this study, a thermal desorption-gas chromatography-mass spectrometry (TD-GC-MS) method following sorbent tube sampling was developed for the determination of monoterpenes (MTs), sesquiterpenes (SQTs) and diterpenes (DTs) in gas-phase samples.*"

We thank the reviewer for pointing out that there was no information about blanks. Most importantly, blank measurements were done, but the information was missing from the original manuscript. This will be addressed more carefully later on here in the answers (under Specific comments)..

ACM: The beginning of Section 2.4 was rewritten and modified:
"Two types of real sample applications were selected in order to test the analytical method performance, to identify compounds emitted by boreal forest tree species and to quantify the emissions from a branch enclosure. The first sample application was a dynamic headspace sampling of compounds emitted by needles/twigs in a closed chamber upon heating, and the second sample application was a study of emissions from a branch enclosure in field conditions. The main emphasis on real sample applications was set on the identification and study of DTs, although MTs and SQTs were also monitored. Based on previous studies, it was considered that since the DTs seem to be emitted from plant storage pools and since temperature has been shown to be a driving force for DTs emissions (Haberstroh et al., 2018;Matsunaga et al., 2012;Yáñez-Serrano et al., 2018), heating of the plant materials would be an efficient way for releasing the DTs into gas-phase. Thus, the studied plant material samples were firstly heated to temperatures ranging from 30 °C to up to 70 °C in order to obtain presumably detectable amounts of DTs to allow their identification and quantification."

In the Section 3.2 the following was added:
"As explained in Section 2.4., in the following experiments the plant material samples were heated in order to characterize the DTs potentially emitted by the studied tree species and to further test the TD-GC-MS method performance in view of real sample analysis. The plant material samples were heated to facilitate the release of DTs from the plant storage pools into the gas-phase."

**3- Method development. The paper needs definitions on method development and validation. Please explain how this is different from previous studies. Furthermore, please discuss while you develop a method for quantification but at the end, for real samples you use a different system and you only give semi-quantitative information.**

AR: The term validation was removed from the manuscript and it was replaced with a more appropriate term analytical figures of merit. This was considered to being more representative for the purpose of this manuscript, although much of the work we conducted falls within the context of method validation. Most of the tests and experiments performed were done for testing and demonstrating that the developed analytical procedure is suitable for its intended purpose, i.e. that the developed method can be used for the analysis of MTs, SQT and most importantly DTs.

In the revised version it has been made clearer how this is different from previous studies (see AR and ACM above to point 1).

The TD-GC-MS2 was used for the other real sample analysis since the TD-GC-MS1 was not available for analysis at the time of these experiments. The TD-GC-MS2 chromatograms from standard and sample analyses are presented in Figs. S8 and S20, respectively. The TD-GC-MS2 was well suited for the analysis of the analytes. The main differences between the TD-GC-MS units are presented in Table S2 and in Section 2.2 (i.e. differences were the models of units, different cold traps and GC columns installed (column stationary phases were the same, but the total length of columns were different and column manufacturers were different). In the revised manuscript, a mentioning about the TD-GC-MS1 not being simply available for analysis was added and a note about column length.

The results are semi-quantitative since the non-target compounds detected in real sample analysis were quantified based on the standard analyte compounds. This is explained in Section 2.5 second paragraph. This paragraph was rephrased and modified in the revised manuscript.

ACM: Please see the ACM above in point 1 to see the changes covering the difference to previous studies. In addition to those, the following changes were made:

Validation was removed and replaced accordingly in all parts of the manuscript.
In the abstract: "The analytical figures of merit were determined, and the method performance was tested by conducting experiments related to, for example, sampling recovery, storage stability and ozone reactivity."
In Section 2.2.: "Most of the analytical method development and performance tests were conducted with TD-GC-MS1"
In Section 2.3.5 title was changed to "2.3.5 Analytical figures of merit"
In Section 3.1.4 title was changed to "3.1.4 Analytical figures of merit of the developed TD-GC-MS method" and in this section the following changes were made "The analytical figures of merit are shown in Table 1."

Regarding semi-quantitative results, in Section 2.5, the second paragraph was modified:
"The non-target compounds detected in real samples were quantified by using a compound from the standard analytes list ($\alpha$-pinene, limonene or linalool were used for MTs and for close eluting BVOCs, $\beta$-caryophyllene or caryophyllene oxide for SQTs and for close eluting BVOCs, and cembrene or ent-kaurene were used for DTs and close eluting BVOCs). Since the non-target compounds were quantified in this manner, the mass concentration results are merely semi-quantitative. A signal-to-noise (S/N) ratio of 10 criteria was used in non-target compound quantification. Similar quantification procedure for non-target compounds has been used also in previous BVOC studies (e.g. Bouvier-Brown et al., 2009; Haberstroh et al., 2018; Hellén et al., 2018)."

Regarding the use of TD-GC-MS2 for analysis of real sample, a comment was added to Section 3.2.2 first paragraph ending: "TD-GC-MS2 was used for analysis since the TD-GC-MS1 was not available at the time of these experiments."

The differences between the units to Section 2.2.: "…The main differences in these TD-GC-MSs were the models of units, different cold traps and columns (same stationary phases, but 10 m shorter column in TD-GC-MS2) installed…"

**Specific comments:**

**Page 1 Line 14: Despite I think ozone reactivity of diterpenes is important and should be stated in the paper, it is really not introduced in the text. . . how these measurements add to your method validations. It must be better introduced here in abstract and in text.**

AR: Ozone is one very important factor causing losses of the studied compounds during the sampling. Therefore it is very important to know how reactive these compounds are and how fast they are lost by the ozone. Some ozone removal methods can be used, but most of them are also prone to cause losses or isomerization of the terpenes. Therefore it is very important to know how long we can sample without using any ozone removal methods, if needed.

ACM: Sentences "Ozone is an important factor causing losses of the studied compounds during sampling. Therefore, losses of terpenes upon ozone exposure were studied and the reaction rate coefficients were estimated" added to the abstract.
And sentence "It is well-known that compounds can be lost during sampling due to reactions with ozone" was added to Section 2.3.6 Ozone reactivity tests.

**Page 1 line 19: I do not understand why heating up to 60_C is representative of anything natural, you will never get these temperatures in boreal ecosystems. In which way these diterpenes will then be emitted to ambient air?**

AR: We did not try to represent specifically real emissions. We simply ended up testing fairly high temperatures in order to characterize the DTs possibly emitted by the studied boreal forest tree species and to test the developed analytical procedure in practice. It was thought to be likely that the diterpenes emitted in the laboratory experiments and proof-of-concept experiments would be the same diterpenes emitted in natural conditions.

It is out of the scope of this study to speculate how the DTs would be emitted in real conditions. Long-term measurement campaigns are needed to study the emissions and only then we can investigate under which real-life conditions the DTs are emitted (if they are emitted at all in detectable levels in boreal forest). This is also stated in the Conclusions. The diterpenes were also emitted at lower temperatures (as can be seen in the Figs. 8 and 9), although in much lower amounts. Nonetheless, we have explained our decision to use high temperatures more clearly in the revised manuscript (essentially the same modification as in major comment point 2).

The decision to test also high temperatures are given in the manuscript later on.

ACM: In Section 2.4:
"Two types of real sample applications were selected in order to test the analytical method performance, to identify compounds emitted by boreal forest tree species and to quantify the emissions from a branch enclosure. The first sample application was a dynamic headspace sampling of compounds emitted by needles/twigs in a closed chamber upon heating, and the second sample application was a study of emissions from a branch enclosure in field conditions. The main emphasis on real sample applications was set on the identification and study of DTs, although MTs and SQTs were also monitored. Based on previous studies, it was considered that since the DTs seem to be emitted from plant storage pools and since temperature has been shown to be a driving force for DTs emissions (Haberstroh et al., 2018;Matsunaga et al., 2012;Yáñez-Serrano et al., 2018), heating of the plant materials would be an efficient way for releasing the DTs into gas-phase. Thus, the studied plant material samples were firstly heated to temperatures ranging from 30 °C to up to 70 °C in order to obtain presumably detectable amounts of DTs to allow their identification and quantification."

In the Section 3.2 the following was added:

"As explained in Section 2.4., in the following experiments the plant material samples were heated in order to characterize the DTs potentially emitted by the studied tree species and to further test the TD-GC-MS method performance in view of real sample analysis. The plant material samples were heated to facilitate the release of DTs from the plant storage pools into the gas-phase."

**Page 1 Line 20: five what?**

AR: Five DTs in the pine needle emissions. For clarification word 'DTs' was added after the five.

ACM: "…, five DTs and 13 DTs could be…"

**Page 1 Line 20: it is really not acceptable that you provide a method validation to then deliver semi-quantitatively.**

AR: We did not have standards for the non-target compounds, hence the emission rate results were presented as semi-quantitative. This is a common practice in non-target analysis, where it is common that the analyst does not possess all the pure compounds for calibration (and quantification). This process is explained in the revised manuscript Section 2.5 in the second paragraph. No changes were made to the Abstract regarding this.

**Page 2 Line 41. The way you express monoterpenoids (as well as sesquiterpenoids and diterpenois) is really confusing. By omitting mt and sqt this problem will be solved. But please do explain what do you mean by monoterpene and monoterpenoid. . . for instance, is p-cymene not a monoterpenoid? And it is only C10H14. . ..In fact, you could name them in Table 1, S4,S5.**

AR: The classification and terminology was updated and these were added in Tables as suggested by the reviewer. This change was also done based on the Reviewer #2 comments. Now it should be hopefully clear to which compounds we refer to since these are classified and shown in all the Tables. The text was revised throughout the manuscript/supplement accordingly. In addition, the title was changed to "…monoterpenes, sesquiterpenes and diterpenes".

ACM: The terminology has been changed throughout the manuscript. As an example, the compound categories were added Section 2.1. and Table 1: "The analyte included nine MTs (both hydrocarbon and oxygenated MTs), six SQTs (both hydrocarbon and oxygenated SQTs), three diterpenes (all hydrocarbon DTs) and three other compounds that were categorised here as BVOCs."
Revised Table 1:

**Table 1. List of analytes, corresponding retention indices (RI) and analytical figures of merit.**

| Compound | CAS Number | Formula | RI[a] | Quan-tify m/z ion | LOQ in pg (in pptv)[b] | Repea-tability (RSD%) | Inter-mediate precision (RSD%) | U[c] (%) | U[d] (%) |
|---|---|---|---|---|---|---|---|---|---|
| α-Pinene (MT) | 7785-70-8 | $C_{10}H_{16}$ | 945 (942) | 93 | 34 (1.2) | 2.6 | 3.8 | 16 | 18 |
| camphene (MT) | 79-92-5 | $C_{10}H_{16}$ | 963 (960) | 93 | 14 (0.5) | 1.6 | 4.0 | 16 | 18 |
| β-pinene (MT) | 19902-08-0 | $C_{10}H_{16}$ | 989 (988) | 93 | 13 (0.5) | 4.2 | 3.2 | 17 | 20 |
| 3-carene (MT) | 498-15-7 | $C_{10}H_{16}$ | 1018 (1017) | 93 | 18 (0.6) | 2.0 | 3.3 | 16 | 17 |
| *p*-cymene (MT) | 99-87-6 | $C_{10}H_{14}$ | 1033 (1031) | 119 | 20 (0.7) | 2.0 | 4.2 | 17 | 19 |
| limonene (MT) | 5989-54-8 | $C_{10}H_{16}$ | 1039 (1037) | 68 | 62 (2.2) | 1.5 | 3.8 | 17 | 17 |
| 1,8-cineol (MT) | 470-82-6 | $C_{10}H_{18}O$ | 1044 (1042) | 154 | 27 (0.9) | 2.6 | 4.7 | 18 | 20 |
| terpinolene (MT) | 586-62-9 | $C_{10}H_{16}$ | 1094 (1093) | 121 | 65 (2.3) | 2.5 | 5.0 | 17 | 18 |
| linalool (MT) | 78-70-6 | $C_{10}H_{18}O$ | 1100 (1100) | 71 | 88 (2.8) | 2.4 | 4.8 | 18 | 18 |
| 4-acetyl-1-methylcyclohexene (4-AMCH) (BVOC) | 6090-09-1 | $C_9H_{14}O$ | 1140 (1139) | 95 | 120 (4.2) | 2.7 | 6.3 | 19 | 20 |
| nopinone (BVOC) | 38651-65-9 | $C_9H_{14}O$ | 1154 (1153) | 83 | 29 (1.0) | 2.7 | 5.3 | 18 | 21 |
| bornylacetate (BVOC) | 5655-61-8 | $C_{12}H_{20}O_2$ | 1295 (1294) | 95 | 49 (1.2) | 2.4 | 6.2 | 20 | 23 |
| longicyclene (SQT) | 1137-12-8 | $C_{15}H_{24}$ | 1401 (1401) | 94 | 35 (0.8) | 2.4 | 4.9 | 18 | 21 |
| isolongifolene (SQT) | 1135-66-6 | $C_{15}H_{24}$ | 1422 (1422) | 161 | 38 (0.9) | 2.5 | 5.7 | 19 | 22 |
| β-caryophyllene (SQT) | 87-44-5 | $C_{15}H_{24}$ | 1445 (1444) | 93 | 105 (2.5) | 2.2 | 3.5 | 17 | 22 |
| β-farnesene (SQT) | 18794-84-8 | $C_{15}H_{24}$ | 1457 (1456) | 69 | 198 (4.7) | 5.0 | 5.9 | 20 | 22 |
| α-humulene (SQT) | 6753-98-6 | $C_{15}H_{24}$ | 1481 (1480) | 93 | 33 (0.8) | 2.1 | 4.2 | 18 | 24 |
| caryophyllene oxide (SQT) | 1139-30-6 | $C_{15}H_{24}O$ | 1612 (1612) | 79 | 340 (7.5) | 18.9 | 8.6 | 24 | 28 |
| cembrene (DT) | 1898-13-1 | $C_{20}H_{32}$ | 1959 (1959) | 93 | 287 (5.1) | 4.0 | 9.5 | 27 | 52 |
| ent-kaurene (DT) | 562-28-7 | $C_{20}H_{32}$ | 2106 (2106) | 257 | 518 (9.3) | 2.9 | 7.7 | 26 | 42 |
| 3-methylene-5-α-androstane (3-MA)[e] (DT) | not assigned | $C_{20}H_{32}$ | 2153 (2152) | 257 | 383 (6.9) | 3.0 | 7.2 | - | - |

[a] RI as Kovats indices (and arithmetic indices in brackets)

[b] LOQ in pptv calculated by using sample volume of 5 L (at T=298 K and p=1 atm)

[c] U (%) for ambient air application

[d] U (%) for branch enclosure application

[e] The purity of this crude standard was set to being 100% (allowing the determination of LOQ, but unknown uncertainty).

**Page 2 line 62. This is confusing, please state which branch enclosure you are referring and which ambient studies? Do you mean you experiments or the previously reported studies?**

AR: This paragraph of the Introduction section was largely modified. To this particular part, the references were added, and the sentences were rephrased.

ACM: "…In general, previously offline (and online) TD-GC-MS technique has been mainly used for the analysis of MTs and SQTs (Hellén et al., 2018;Mermet et al., 2019; Pankow et al., 2012), but the above-mentioned branch enclosure and ambient air studies indicate that this analytical technique is also applicable for the analysis of DTs (Haberstroh et al., 2018;Matsunaga et al., 2012;Yáñez-Serrano et al., 2018). Unfortunately, most of those studies did not present any detailed method descriptive performance parameters for DTs (e.g. detection limits, reproducibility, selectivity or sampling recovery), making it difficult to critically evaluate the suitability of the analytical procedures for the analysis of low volatility DTs (Haberstroh et al., 2018;Matsunaga et al., 2012;Yáñez-Serrano et al., 2018)…"

**Page 2 Line 63. If you are going to say this, you need to explain what do you mean by method validation. Do you need two different analytical systems to prove your concentrations? And what do you mean by performance parameters? GC and sampling settings? In is normally well mentioned in the methods of each study? Please state why you method is 1) better than previously mentioned for DT measurements, and 2) how you truly validate the method rather than say your method is good because I do not loose DT in the lines. . .**

AR: This paragraph of the Introduction section was modified.

In those referred DT publications, two out of six publications gave only limit of detection values for DTs, the other four either do not mention detection limits or if they do, those are only for MTs and SQTs. Information about, for example, calibration range, repeatability, reproducibility, selectivity, uncertainty or recovery concerning DTs is not directly presented in most of the DT studies. This is what we tried to discreetly say in the Introduction section. In principle, analytical procedures used for MTs and SQTs (and other BVOCs or VOCs) have been applied to analyze DTs, but there is no separate information presented that shows that the methods are reliable for that purpose. Of course, it could be argued that the ambient results indicate that the methods have worked, but strictly speaking, since there are no experimental tests showing clearly the performance parameters, there is a justified reason for questioning the applicability of such analytical procedures for DT analysis. In many of those studies, there have been statements that the concentrations of DTs might be underestimated due to sampling line losses, however, the recoveries of DTs have not been separately tested.

To the point 1): In the manuscript, we do not declare that our method is better than the previously used methods for the analysis of DTs in gas-phase samples. Actually, it is quite difficult to compare our method to the ones already existing, since as explained above, those methods unfortunately do not have that much information about DTs presented to compare to. In the manuscript Section 3.1.4, we compare our LOQ values to the detection limits presented in one prior publication (the one publication that showed detection limits in comparable units for DTs). However, no changes regarding this were made to the Introduction section.

To the point 2): The term validation was removed from the manuscript and it was replaced with a more appropriate term analytical figures of merit. This was considered to being more representative for the purpose of the manuscript, although much of the work we conducted falls within the context of method validation. We hope this terminological change satisfies the reviewers' concerns related to

method validation. In general, the experiments performed to demonstrate that the method is suitable for its intended purpose included, for example: selectivity and specificity (showed chromatographic separation, compounds identified with mass spectrometry), repeatability (done six repetitions of a standard sample during one day), intermediate precision (done 22 repetitions of a standard sample during 4.5 months period), limit-of-quantification (determined by analyzing 10 blanks), calibration ranges, recoveries (different inlet sampling lines tested), compound stability during storage (tested different storage times), estimation of measurement uncertainty and finally the real sample application tests.

ACM:
In the Introduction section, one paragraph was largely rewritten:
"Diterpenes have fairly low vapor pressures and they can be categorised as semi-or low-volatile organic compounds. The determination of DTs in atmospheric gas-phase samples can be challenging due to their low-volatility and high reactivity, which might cause sampling line losses and emphasize the need for high-sensitivity detection methods (Yáñez-Serrano et al., 2018;Yee et al., 2018). In previous atmospheric studies, DTs have been analysed by using either conventional chromatographic techniques or online mass spectrometric techniques (such as proton-transfer-reaction mass spectrometry) (Chan et al., 2016;Haberstroh et al., 2018;Li et al., 2019;Matsunaga et al., 2012;Yáñez-Serrano et al., 2018;Yee et al., 2018). Many of those chromatographic studies targeting DTs used traditional sorbent tubes for sampling prior to thermal desorption (or solvent desorption)-gas chromatography-mass spectrometry (TD-GC-MS) analysis (Haberstroh et al., 2018;Matsunaga et al., 2012;Yáñez-Serrano et al., 2018). In general, previously offline (and online) TD-GC-MS technique has been mainly used for the analysis of MTs and SQTs (Hellén et al., 2018;Mermet et al., 2019; Pankow et al., 2012), but the above-mentioned branch enclosure and ambient air studies indicate that this analytical technique is also applicable for the analysis of DTs (Haberstroh et al., 2018;Matsunaga et al., 2012;Yáñez-Serrano et al., 2018). Unfortunately, most of those studies did not present any detailed method descriptive performance parameters for DTs (e.g. detection limits, reproducibility, selectivity or sampling recovery), making it difficult to critically evaluate the suitability of the analytical procedures for the analysis of low volatility DTs (Haberstroh et al., 2018;Matsunaga et al., 2012;Yáñez-Serrano et al., 2018). In general, sorbent tube sampling followed by TD-GC-MS analysis methods have been demonstrated to be suitable for the analysis of MTs, however, there is very little detailed information about the suitability for the analysis of SQTs (Bouvier-Brown et al., 2009;Helmig et al., 2004;Jones et al., 2014;Mermet et al., 2019;Pankow et al., 2012), and data related to DTs is currently lacking"

In the Introduction section last paragraph, the following clarification was done: "In this work, we build on our previous knowledge of the TD-GC-MS technique and developed our core TD-GC-MS method further to allow the analysis of the fairly low-volatility DTs. In this study, the method development and applicability related experiments done in view for the analysis of DTs included e.g. desorption efficiency, sampling recovery, stability tests, comparison of online and offline sampling modes and ozone reactivity tests. The method analytical figures of merit were also determined, including the quantification limits, intermediate precisions and measurement uncertainties for the terpenes. In addition, qualitative and semi-quantitative experiments were conducted in laboratory and in field conditions to characterise the DTs potentially emitted by boreal forest tree species."

Validation was removed and replaced accordingly in all parts of the manuscript as follows:
In the abstract: "The analytical figures of merit were determined, and the method performance was tested by conducting experiments related to, for example, sampling recovery, storage stability and ozone reactivity."

In Section 2.2.: "Most of the analytical method development and performance tests were conducted with TD-GC-MS1"

In Section 2.3.5 title was changed to "2.3.5 Analytical figures of merit"

In Section 3.1.4 title was changed to "3.1.4 Analytical figures of merit of the developed TD-GC-MS method" and in this section the following changes were made "The analytical figures of merit are shown in Table 1."

**Page 3 Line 69-78. In your objectives are somehow confusing. First you talk about the applicability of sorbent tubes and GCMS for mt, sqt and dt. I do not know what is new here. This has been shown in the past, plus this is no objective, but rather telling what you did. Later you say the method was incorporated from previous studies for sqt and mt. Thus there is no novelty on sqt and mt, so this builds my point of removing mts and sqt from the text. Then you say you do a method development for dt, and it is not clear to me what is the actual development (not just simply that you have calculated the LOD. . .. But rather than the lod is lower than using this other method. . ..) as compared to mt and sqt analysis. It is also very strange to me that you offer a validated method and then you report only semi-quantitative values. Thus the whole objectives section must be rewritten. Please bear in mind to put what do you intent to do and why.**

AR: Please see the AR's and ACM's to main comments 1-3. This objectives paragraph was rewritten partly as suggested. Some of the results were semi-quantitative since we did not have pure standards for the non-target compounds.

ACM: The last paragraph of Introduction was largely rephrased and rewritten:
"In this study, sorbent tube sampling followed by TD-GC-MS analysis for the determination of MTs, SQTs and especially DTs in gas-phase samples was developed and evaluated. The TD-GC-MS method was partially incorporated from our group's previous studies (see e.g. Hellén et al., 2018), and modified accordingly to fit the needs for the analysis of low volatility diterpenes. In principle, similar analytical procedures have been used previously in both ambient air and enclosure emission studies for the analysis of relatively volatile and semi-volatile MTs and SQTs in multiple field campaigns (Aaltonen et al., 2011;Hellén et al., 2018;Joensuu et al., 2016;Mäki et al., 2017;Vanhatalo et al., 2018). In this work, we build on our previous knowledge of the TD-GC-MS technique and developed our core TD-GC-MS method further to allow the analysis of the fairly low-volatility DTs. In this study, the method development and applicability related experiments done in view for the analysis of DTs included e.g. desorption efficiency, sampling recovery, stability tests, comparison of online and offline sampling modes and ozone reactivity tests. The method analytical figures of merit were also determined, including the quantification limits, intermediate precisions and measurement uncertainties for the terpenes. In addition, qualitative and semi-quantitative experiments were conducted in laboratory and in field conditions to characterise the DTs potentially emitted by boreal forest tree species."

**Page 3 Line 83: If one of the dt standard is not analytically valid, then you have to skip it.**

AR: It was written "not analytically valid in terms of purity and traceability". By purity, it was meant that the supplier did not provide purity of the compound, it was merely a crude standard. By traceability, it was meant that no CAS number is assigned to the compound. However, the compound was purchased from a major supplier (Sigma-Aldrich, Table S1) and should be available for others if they are interested in reproducing our analytical procedure. We merely wanted to be transparent in the manuscript so that reader is aware of this limitation of the chemical. In our opinion, there is no apparent reason for removing the results of this compound.

**Page 3 Line 84-86. Delete sentence**

AR: In this sentence, we elaborate the reason for why we had only three DTs as standards and also emphasize the reason for including MTs and DTs as analytes. We prefer to keep this sentence in the manuscript, however, the sentence was modified a bit to make the reasoning clearer.

ACM: "The current supply situation of DTs is not optimal for the purpose of comprehensive method development, however, together with the results of MTs and SQTs to compare to, valuable information was expected be obtained even with only few DTs included as analytes."

**Page 3 Line 88: Please state why do you dilute the standards in methanol and not other solvent such as hexane.**

AR: Most importantly, all the analytes are soluble in methanol. Also, methanol is a good solvent in practice for the calibration setup, since it volatilizes easily in the nitrogen stream. Other less volatile solvents might retain in the sorbent and be observable in relatively large amounts in the chromatograms. This was added in Section 2.1 together with a reference to previous work.

ACM: In Section 2.1 last paragraph: "Calibration tubes were prepared by injecting 5 µL of individual standard solution into the tube and simultaneously applying 80 mL min$^{-1}$ flow of nitrogen (≥99.9999 % AGA, Espoo, Finland). The tubes were purged for 10 min in order to evaporate the excess methanol (Hakola et al., 2003)."

**Page 3 line 88 and 90. You can't report such a large range in concentrations for analyte solutions. You must be certain of the concentration of your solution, and this is why you use calibrated pipettes. So please change accordingly. If you have a range of concentrations due to different compounds, then state them in a table.**

AR: Indeed we do have different concentrations for different compounds. Individual analytes were weighted in the same flask, thus the analytes were mixed in the same standard solutions. The ranges are expressed for different compound categories later on in the manuscript (see the answer to second following comment from this comment), however, we also rephrased slightly this Section 2.1 second paragraph. It was added that the solutions were a mixture of compounds and that the range expressed depended on the compound.

ACM: In section 2.1: "Primary standard mix solutions of the target compounds were prepared by weighting and diluting the pure compounds in methanol. The analyte concentrations in primary solution were in the range of 40–200 mg L$^{-1}$ depending on the compound."

**Page 3 line 91. Why do you store solutions in the dark at 4_C?**

AR: Some of the pure analytes are recommended to be stored at 2-8 °C (e.g. α-pinene, α-humulene and ent-kaurene), thus also the standards were stored at 4 °C. They were mostly "in the dark", since the refrigerator light turns off when the door is closed. No changes were made to the manuscript.

**Page 4 line 103-104. I do not understand this sentence. Please rephrase thinking what is the point of this statement to the objective of the paper.**

AR: The point is to show the calibration standard concentration ranges. The sentence was rephrased by adding "…calibration concentration ranges…" and by decreasing the number of decimals shown and putting the "expressed as absolute amount (ng) in sorbent tube" at the end in parenthesis.

ACM: "The calibration concentration ranges were 0.2–55.8 ng for MTs, 0.2–61.2 ng for SQTs, 0.4–200 ng for DTs and 0.2–43.2 ng for other BVOCs (expressed as absolute amount (ng) in sorbent tube)."

**Page 5 Line 130: why you changed the method (different final state) for TDGCMS2?**

AR: The column (different manufacturer, but same stationary phase) in TDGCMS2 was 10 m shorter than the column in TDGCMS1 (Table S1). Also, it was considered that it is beneficial for the column lifetime to shorten the final state hold temperature. It did not change the separation of the compounds. A separate mention about the column length was added to earlier part in Section 2.2.

ACM: "…different cold traps and columns (same stationary phases, but 10 m shorter column in TD-GC-MS2) installed."

**Page 5 Line 136-140: I think this information is useless, either rephrase or remove.**

AR: These lines were removed from this Section 2.3, however, parts of the original information was modified and added to the beginning of Section 2.2.

ACM: "As mentioned before in the Introduction, the TD-GC-MS method used here was partially applied from a previously developed and validated in-house method that has been used in our group's previous studies for the analysis of MTs and SQTs (see e.g. Hellén et al., 2012, 2018). In this current work, the analytical GC column stationary phase (nonpolar 1,4-bis(dimethylsiloxy)phenylene polydimethylsiloxane) film thickness was decreased from a previously used thick phase 1.0 μm column to 0.25 μm column as a compromise to maintain proper selectivity between the terpenes and to elute the DTs in reasonable time (chromatograms presented later on). It was expected that the DTs might retain too strongly in the 1.0 μm film thickness column and that the column bleed could become a problem in long-term use since high temperatures are required to elute the DTs. Thus, the analytical column was replaced, GC oven temperature program was modified, and the MS scan settings were updated and modified. For the most part, the TD unit parameters' were kept as in the previously optimized in-house method, however, the desorption efficiency was separately tested and verified in this work."

**Page 5 Line 146: please state how many tubes were stored.**

AR: Done.

ACM: "…was evaluated by storing the tubes (*n*=2-6) sealed…"

**Page 6 line 186: please state which are the guidelines so the reader doesn't need to go somewhere else to find it. You can do it in SI if you don't want this info in the text.**

AR: This was done as the reviewer suggested by showing the formulas in Supplementary material. We also improved the presentation of the measurement uncertainties by showing the U values for two applications in which the developed analytical procedure could be used: i) offline sorbent tube sampling of ambient air followed by TD-GC-MS analysis and ii) offline sorbent tube sampling of emissions from a branch enclosure cuvette followed by TD-GC-MS analysis.

ACM: The sentence was changed: "The expanded measurement uncertainty (U) was estimated from partial uncertainties by following ACTRIS (Aerosol Clouds Trace gases Research InfraStructure) guidelines (ACTRIS, 2018). The U (%) was estimated for two analytical procedures: i) offline sorbent tube sampling of ambient air followed by TD-GC-MS analysis and ii) offline sorbent tube sampling of

emissions from a branch enclosure cuvette followed by TD-GC-MS analysis. The intermediate precision and recovery experiment results were included in the calculations of U. In the ambient air sampling procedure, the ozone removal inlet recovery results were used in the calculations, whereas in the branch enclosure emissions procedure the cuvette recovery results were used in the calculations. A more detailed description of the uncertainty calculations can be found in Supplement Text S1."

In supplement the following has been added:

"TEXT S1. Expanded measurement uncertainty

The expanded measurement uncertainty (U) was estimated from partial uncertainties of the procedures by following ACTRIS (Aerosol Clouds Trace gases Research InfraStructure) guidelines (ACTRIS, 2018). The U was estimated separately for two analytical procedures: i) offline sorbent tube sampling of ambient air followed by TD-GC-MS analysis and ii) offline sorbent tube sampling of emissions from a branch enclosure cuvette followed by TD-GC-MS analysis.

The combined standard uncertainty (total uncertainty, $u\mathcal{X}_{\text{total}}$) includes both the random and systematic errors affecting the measurements. In this work, the $u\mathcal{X}_{\text{total}}$ was calculated for both procedures i) and ii) by using Eq. (S1):

$$u\mathcal{X}_{\text{total}}^2 = u\mathcal{X}_{\text{prec}}^2 + u\mathcal{X}_{\text{stdprep}}^2 + u\mathcal{X}_{\text{vol}}^2 + u\mathcal{X}_{\text{rec}}^2 \ , \qquad\qquad\qquad (S1)$$

where $u\mathcal{X}_{\text{prec}}$ is the precision component taking into account random errors, $u\mathcal{X}_{\text{stdprep}}$ is the uncertainty due to calibration standard preparation, $u\mathcal{X}_{\text{vol}}$ is the component descriptive of systematic errors in the sample volume determination and $u\mathcal{X}_{\text{rec}}$ is the recovery component taking into account the losses in sampling lines and/or cuvette.

The $u\mathcal{X}_{\text{prec}}$ was calculated by following Eq. (S2):

$$u\mathcal{X}_{\text{prec}}^2 = \left(\mathcal{X}_{\text{sample}} * \sigma_{\text{series}}^{rel}\right)^2 + \left(\frac{x_{\text{LOD}}}{3}\right)^2 \ , \qquad\qquad\qquad (S2)$$

where $\mathcal{X}_{\text{sample}}$ is the concentration of the analyte, $\sigma_{\text{series}}^{rel}$ is the intermediate precision expressed as relative standard deviation and $x_{\text{LOD}}$ is the limit of detection (LOD) value. The intermediate precision was obtained from analysing standard samples ($n$=22) during 4.5 months period (Section 2.3.5). The $x_{\text{LOD}}$ was determined by analysing multiple blank sorbent tubes (n=10), and then by calculating the standard deviation of peak area in blanks and by multiplying the standard deviation by three.

The $u\mathcal{X}_{\text{stdprep}}$, was derived by taking into account the uncertainties originating from weighting (mass balance error), pipetting (volumetric pipette error) and diluting (volumetric flask error) the standard compounds. The $u\mathcal{X}_{\text{stdprep}}$ was eventually calculated by setting a 5% uncertainty, as in Hellen et al. (2002), for all the analytes included in this work. This was considered as an upper estimate for the calibration standard preparation uncertainty.

The $u\mathcal{X}_{\text{vol}}$ was estimated based on the uncertainty of the offline sampling by estimating the collective uncertainty related to sampling flow rate and sampling time. In this work, the $u\mathcal{X}_{\text{vol}}$ for all compounds was derived by using a set 5% uncertainty. This was considered as being an upper estimate for the sampling volume uncertainty.

The $u\mathcal{X}_{\text{rec}}^2$ was calculated based on the sampling line recovery results by using Eq. (S3):

$$u\mathcal{X}_{\text{rec}}^2 = \left(\mathcal{X}_{\text{sample}} * \sqrt{\frac{\sum (bias)^2}{n}}\right)^2, \qquad\qquad\qquad (S3)$$

where $\mathcal{X}_{sample}$ is the concentration of the analyte, *bias* is the deviation of the observed relative recovery from the 100% recovery and $n$ is the number of different measurements. The recovery results obtained with the 1 m long ozone removal inlet at two different relative humidity (RH) levels (RH=0% and RH=100%) were used for the procedure i), whereas the recovery results obtained in the cuvette experiments at flow rates 2.0 L/min and 6.7 L/min were used for the procedure ii) (see Sections 2.3.3 and 3.1.3). These were considered to being representative of real applications. In procedure i), ambient air can be sampled directly via short ozone removal inlet (as in e.g. Hellen et al., 2018). In procedure ii), the sorbent tubes are sampled from the cuvette outlet port, therefore, there is no need for long sampling lines and ozone removal inlet is not needed since the incoming air is zero VOC free air (as in e.g. Hakola et al., 2006).

Finally, the expanded measurement uncertainty (U) was calculated by using Eq. (S4):

$$U = 2 * u\mathcal{X}_{total} = 2 * \left( \frac{u\mathcal{X}_{total}}{\mathcal{X}_{sample}} * 100\% \right). \tag{S4}$$

A coverage factor of k=2 was used for providing a level of confidence of 95 % and the uncertainty was finally converted to relative amount to expresses the U in percentage. The expanded measurement uncertainties for both procedures are presented in the main text Table 1.

**References**

ACTRIS, 2018. Deliverable 3.17. Updated Measurement Guideline for NOx and VOCs, available at: https://www.actris.eu/Portals/46/Documentation/actris2/Deliverables/public/WP3_D3.17_M42.pdf?ver=2018-11-12-143115-077. (last access:27 March 2020).

Hellén, H., Hakola, H., Laurila, T., Hiltunen, V., and Koskentalo, T.: Aromatic hydrocarbon and methyl tert-butyl ether measurements in ambient air of Helsinki (Finland) using diffusive samplers, Sci. Tot. Environ., 298, 55-64, 2002.

Hellén, H., Praplan, A. P., Tykkä, T., Ylivinkka, I., Vakkari, V., Bäck, J., Petäjä, T., Kulmala, M., and Hakola, H.: Long-term measurements of volatile organic compounds highlight the importance of sesquiterpenes for the atmospheric chemistry of a boreal forest, Atmos. Chem. Phys., 18, 13839-13863, doi:10.5194/acp-18-13839-2018, 2018.

Hakola, H., Tarvainen, V., Bäck, J., Ranta, H., Bonn, B., Rinne, J., and Kulmala, M.: Seasonal variation of mono- and sesquiterpene emission rates of Scots pine, Biogeosciences, 3, 93-101, doi:10.5194/bg-3-93-2006, 2006. "

**Page 7 line 206: at what temperature did you store it? Then you freeze the material? is not clear how you did it.**

AR: There was a wrong word in this sentence that made no sense for the reader. It should have been written *freezer* instead of *refrigerator*. In order to avoid any further confusion, the wording was replaced with temperature.

ACM: "The samples were stored in the dark at -18 °C for a maximum of 2 months prior to analysis."

**Page 7 line 214: what do you mean as a proof-of-concept type of experiment? And please state if the branch enclosure was just a branch or it also had needles in.**

AR: The branch had needles. This information was added.

Proof-of-concept type of experiment refers here to a design of a experiment that demonstrates that the developed method works in a practical application, i.e. that the analytical procedure is suitable for the purpose it was developed. This was added to the revised version of the manuscript.

ACM: The following corrections were done:
"Spruce (Picea abies) branch with needles was placed inside…"
"Branch enclosure emissions were studied on-site at the SMEAR II station as a proof-of-concept type of experiment on two different days (i.e. to test and to demonstrate that the analytical procedure is suitable for the purpose it was intended for)."

**Page 7 line 215: What do you mean but shrank?**

AR: It was meant that it was the same cuvette (Teflon bag) as in Section 2.3.3, but that the volume of the bag was made smaller (from 6.2 L to 1.2 L). This part of the sentence in parenthesis was removed since the information was not really necessary.

ACM: "Spruce (Picea abies) branch with needles was placed inside a ca. 1.2 L Teflon bag cuvette, which was connected to a fixed plate with inlet and outlet ports."

**Page 7 line 217: state reason for copper tube.**

AR: Copper tube was used for improving the heat transfer, if that is what the referee is meaning with this. Any other metal tubing could have worked, but copper was easily available to us. This was made clearer.

ACM: "…and passed through a ca. 1 m long copper tube (o.d. 12 mm) wrapped with a heating wire and insulator cover (copper tubing was used for improving heat transfer). The incoming air was heated to increase the cuvette inside temperature…."

**Page 7 line 219: what is the reason of artificially promote BVOC emissions? Why would you do this if it is not realistic to the ecosystem of interest.**

AR: Maybe the wording *artificially* is a bit awkward in this context, but as it was said earlier in the paragraph, this was a proof-of-concept type of experiment. The point was to test the method in practice with a similar type of setup that would be employed in a conventional monitoring campaign. The decision to use heating came after the laboratory experiments (i.e. after the results presented in Section 3.2.1). However, this part of the sentence was rephrased by removing "artificially" and by replacing BVOCs with DTs, since the decision to use heating was related specifically to DTs.

ACM: "The incoming air was heated to increase the cuvette inside temperature in order to promote the emissions of DTs."

**Page 7 application to real samples. I do not see any blank measurements, please include them. If you have not taken blank measurements then I ask how do you exclude contamination effects.**

AR: Thank you for pointing this out. Blanks were done always before the real samples were collected, but in the original manuscript this information was missing. The blank levels were so low (target/non-target compounds were absent in them) that there was no need for blank subtraction. In the real sample applications, blank samples were used to check possible contaminant peaks from real sample peaks. It was a mistake not to include this information in the original manuscript (it was simply forgotten since no blank subtraction was needed but blanks' information should have been included

since blanks were used to check that the peaks were speciated correctly). This information has been now added to the manuscript.

ACM: Information was added to Section 2.4:
"Blank samples (empty chamber heated) were collected prior to experiments."
"Blank samples (empty cuvette heated) were collected prior to experiment."

Information was added to Section 2.5:
"The analysed blank samples were used to exclude possible contaminant peaks."
"No blank subtraction was needed in the calculations since most analytes were absent in the blank samples."

Blank chromatograms were added to Figs. 6 and 7 (old supplementary Figs. S19 and S20):

[Figure]

"Figure 6. Selected ion recording (SIR) chromatograms from the TD-GC-MS1 analysis of pine needles and blank dynamic headspace sorbent tube samples (sampling at 60 °C for 20 min at a flow rate of 100 mL/min)…"

[Figure]

"Figure 7. Selected ion recording (SIR) chromatograms from the TD-GC-MS1 analysis of spruce twigs and blank dynamic headspace sorbent tube samples (sampling at 60 °C for 20 min at flow rate of 100 mL/min)…"

**Page 8 line 231: what is the sufficient confidence?**

AR: As stated in the sentence, if the RI difference was too high or if the mass spectra did not match between the non-target compound and the proposed molecule, the peak could not be tentatively identified. This was made more clearer in the revised version and the sufficient confidence part was removed.

ACM: "If an unknown compound could not be tentatively identified (either too high RI difference or not adequately matching mass spectra), a proposed compound formula was deduced based on the mass spectra.."

**Page 8 line 233-5: if you can't identify the compounds then remove them from identified compounds.**

AR: They were considered to be *tentatively identified* since no pure standards of the non-target compounds were available. Hence, it was emphasized that the compounds were not *absolutely identified*. To clarify, for example in the pine needle emissions rimuene was tentatively identified based on both the mass spectra and RI matches, but since we did not have rimuene as pure compound, we cannot declare it to be absolutely identified. This is a common procedure in non-target analysis, the compounds are declared to be tentatively identified. This was corrected and made clearer. References were added to support the procedure used here. In addition to changes made in Section 2.5, the terminology used in Section 3.2 was checked and revised accordingly.

ACM: In Section 2.5 the first paragraph was rewritten:
"The non-target compounds in real sample applications were tentatively identified based on both retention index (RI) values and mass spectra comparison to NIST mass spectral library and/or Adams (2007) library. The n-alkane based RI values were calculated based on both Kovats Index (KI) and Arithmetic Index (AI) as shown in Adams (2007). If an unknown compound could not be tentatively identified (either too high RI difference or not adequately matching mass spectra), a proposed compound formula was deduced based on the mass spectra. The non-target compounds that were either tentatively identified or assigned with a formula were then categorised as MTs, SQT and DTs when applicable, the others were categorised as BVOCs. For example, if the non-target compound had the characteristic m/z 272, 257, 243 and 229 ions of DTs and the RI value was representative of DTs (Adams, 2007), the compound was assigned in this study to the plausible DT category. Similar identification procedure of non-target compounds has been used also in previous BVOC studies (e.g. Chan et al., 2016; Hellén et al., 2018; Kännaste et al., 2013; Yee et al., 2018). The compounds were marked here as tentatively identified since authentic standards were not available, thus the final verification is lacking. The analysed blank samples were used to exclude possible contaminant peaks."

In Section 3.2. in three places the wording "positively" was replaced with "tentatively", since even though cembrene (one of our standard analytes) was among the compounds identified in emissions, the others cannot be declared positively identified. The term "unidentified" was changed to "unknown" in the revised Figure captions.

**Page 8 line 241: What is C1?**

AR: This should have been $C_{in}$ as marked in the emission rate equation. Now corrected.

ACM: "In both real sample applications, the $C_{in}$ was negligible…"

**Page 8 line 348-9. Remove this sentence, it does not contribute anything to the ms.**

AR: Done.

ACM: Sentence removed.

**Page 9 line 250: what is the desorption efficiency of TDGCMS2?**

AR: Information was missing, this was added.

ACM: "With the TD-GC-MS2 desorption time needed to be increased from 1 min to 5 min at 300 °C (both sorbent and cold trap) to minimise the carryover, which arise either from the sorbent tube or cold trap, and to obtain sufficient desorption efficiency (≥99.7%)."

**Page 9 line 251: what do you mean by carryover?**

AR: Here it is meant either the carryover from the sorbent tube or from the other parts of the instrument, likely the cold trap. This information was added.

ACM: "…to minimise the carryover, which arise either from the sorbent tube or cold trap, and to obtain sufficient…"

**Page 9 line 269: why did you choose 60 and not 70 or 50C? How did you know all liquid goes into the gas phase?**

AR: This temperature (60 °C) was considered and later observed to being adequate for improving the volatilization of DTs. We didn't test higher temperatures in order to avoid possible PTFE contamination peaks. A comment was added related to this.

As it was explained in the manuscript, it was not a requirement that all the analytes are volatilized, since the tests were performed by relative comparison (sampling sorbent tubes before and after the studied setup, Fig. S1). However, as can be seen in Fig. 2a, the theoretical and observed amounts agreed fairly well, thus with most of the compounds the volatilization was close to "complete". This was clarified in the revised version.

ACM: The first paragraph of Section 3.1.3 was revised:
"In Fig. 2a are shown the amount of each compound detected in the first sorbent tube (after the mixing line, Fig. S1a) and the expected theoretical amount calculated based on the experimental conditions used. As can be seen in Fig. 2a, the observed amounts were mostly in close agreement with the expected amounts. Although this was not a prerequisite for the recovery experiments, the relative recovery was calculated based on amounts found in tubes before and after the sampling line, it demonstrates that the experimental setup was working adequately in practice (although the reproducibility was a bit poor for some compounds, see Fig. 2a). The injection t-piece needed to be heated to 60 °C in order to increase the volatilisation of the diterpenes. If there was no heating applied, then e.g. ent-kaurene yield in the first tube was less than 20 % of the expected amount (data not shown). The diterpenes were observed to stick into the walls of the injection PTFE t-piece if it was not heated, thus heating is recommended to be used in similar experiments. Higher temperatures were not tested here in order to avoid the possible PTFE contamination peaks."

**Page 9 line 271: how did you observe that diterpenes were sticking into walls and not decomposing some other way?**

AR: This was observed in the preliminary tests and in operational error situations. For example, in the preliminary tests we did not use heating of the injection t-piece. Afterwards when heating was applied, the amounts of DTs detected in the first tests were in great excess demonstrating the sticking into the walls (the DTs that had not volatilized when the t-piece was not heated were now volatilized). In addition, the fact that the recoveries were good indicate that the DTs were not decomposing in any other ways. Another clue that indicates that the DTs were not decomposed in other ways was that no decomposition products were detected in the chromatograms where otherwise DTs were absent.

Since this information is not really required to be explained in the manuscript and it would disturb the flow of the text, no changes were made.

**Page 9 line 273-279: This is a very serious statement which I think is wrong. It is fine that you did not find any losses with you inlet line, but this will be highly dependent upon the ambient temperature that the line will be subjected to. For instance 15 m non heated Teflon line for outdoor sqt monitoring in a boreal ecosystem during winter HAS to be heated!. Please remove statement about not heating the Teflon tube, it is totally misleading.**

AR: Yes, the referee is absolutely right that this statement was too generalizing. It was not our intention to mean that the non-heated line could be used e.g. in cold season measurements. This part related to field measurement of the sentence was removed. The remaining sentence was modified.

ACM: "The results indicate that under the applied conditions the Teflon FEP sampling lines do not need to be heated excessively above room temperature (22±1 °C)."

**Page 9-10 lines 280-297: I wonder what will be the effect of transpiration inside a cuvette. This must be really important for diterpenes as you mention, so I think it is wise to let people know that inside cuvette humidity monitoring is essential.**

AR: We added recommendation on humidity monitoring later in Section 3.1.5.

ACM: A sentence on this was added: "Since the RH was shown to have an effect on the recoveries, humidity monitoring of the sample air stream is recommended."

**Page 10 line 288-290: why do you think there is a longer residence time in the cuvette for diterpenes? I think this whole paragraph needs to be rewritten. You simply say that you may not be volatilizing well your diterpenes, thus, you can't be really validating the method if you are not sure you can't get all your diterpenes into the gas phase. Please rephrase by 1) taking away the mt and sqt info (because as you say it is explained in other studies) 2) you think account for this HUGE limitation and how this affects your method validation. In fact, somewhere in the text you must state why you are truly validating a method here with these limitations.**

AR: It is not stated that the residence time is longer only for DTs. The residence time inside the cuvette is longer for all the compounds at 1 L/min flow rate than at 6.7 L/min flow rate. It was meant that since the recoveries of DTs were worse at 1 L/min flow rate (recovery=44%) than at 6.7 L/min flow rate (recovery=76%), likely the residence time inside the cuvette was too long for these low volatility compounds. The recoveries of MTs and SQTs did not change much between the different flow rates

tested. (Please note that this is an example of results where the information obtained based on MTs and SQTs is very useful for interpreting the DTs results.)

As it was explained in the manuscript, it was not a requirement that all the analytes are volatilized completely since the recoveries were calculated by comparing the concentrations observed before and after the studied setup (in this case a cuvette, Fig. S1c). The problem of volatilization is not in our analytical method, it is in the experimental setup used for testing and it is caused by the physicochemical properties of the target analytes.

To clarify, the problem is this cuvette recovery testing setup was two-phase:
- If high flow rate was used, the heating of the injection t-piece was not efficient enough to maintain the same volatilization efficiency as with the low flow rates. Thus, much lower than expected amounts were detected (in the sorbent tube prior to cuvette, amount shown in Fig. S10)
- If low flow rate was used, the volatilization of DTs was adequate (the amount observed in the sorbent tube prior to cuvette was closer to expected amount), but the recoveries of DTs were poor.

To make it clearer, we raised the cuvette recovery results figure from the supplementary information to the manuscript. This new Fig. 3 in the revised manuscript likely helps to follow the results more easily and will help the reader in interpreting the results. In addition, the paragraph was modified and rephrased.

ACM: Figure was raised from the supplementary information to help the reader, now it is Fig. 3.
Section 3.1.3 last paragraph modified/rephrased:
"The cuvette recovery results showed some variability (Fig. 3). As mentioned in Sect. 2.3.3, three different flow rates were tested (inlet flows 6.7, 2.0 and 1.0 L min-1). The recovery values were acceptable at all flow rates tested for all other compounds except for the diterpenes (Fig. 3). The recoveries of MTs and SQTs were on average 95±1 % and 94±2 % at 1.0 L min-1 and 96±1 % and 93±3 % at 6.7 L min-1 flow rates, respectively. However, the recoveries of DTs were on average 44±11 %, 80±6 % and 76±4 % at flow rates of 1.0, 2.0 and 6.7 L min-1. The poor recoveries of DTs observed at the lowest flow rate tested indicate that the residence time inside the cuvette was likely too long for these low volatility compounds (note that for MTs and SQTs the recoveries were acceptable). The DT recoveries were seemingly adequate with the other flow rates tested, indicating that the set flow rate range of ca. 2–7 L min-1 would be suitable for sampling. However, with the highest flow rate tested, the absolute mass concentrations of DTs were very close to LOQ values (Fig. 3). This was likely related to an experimental problem in the analyte volatilisation setup. The injection t-piece heating was not efficient enough for allowing complete volatilisation of the DTs, since the high flow rate rapidly cooled the t-piece. Therefore, these DT cuvette recoveries need to be considered as tentative results, since the concentrations levels were not systematic between the different flow rates tested (Fig. 3). A better sample introduction system would be needed to get the volatilisation more systematic at all flow rates under investigation."

**Page 10 line 299. I do not understand how you say first that you can't really be sure of 3MA and then you actually show LOQ values. How is this possible?. Please explain.**

AR: This is a good question and something that we struggled a bit with during the experimental planning of this work. As it was shown in Table S1, the 3-MA was a crude standard (purity documentation was separately asked from the supplier, but apparently it has not been determined for this chemical). Since the purity was not declared, we simply assumed that all that was weighted in primary standard was pure compound. Thus, we can state the LOQ, but the uncertainty is unknown.

Including the LOQ was considered to be useful information for comparative purposes (both for others planning to reproduce our work and for comparison to MT and SQT results).

ACM: In the revised manuscript in revised Table 1, a footnote was added to 3-MA and in that it was stated: "The purity of this crude standard was set to being 100% (allowing the determination of LOQ, but unknown uncertainty).

**Page 10 line 300: please state that RSD means relative standard error. I wonder why you did not show any repeatability among different systems. This will really validate your method!**

AR: Error bars are shown in all the plots presented in the manuscript, and in those plots the error bars correspond to either repeatability or reproducibility (these are already stated in the figure captions). Intermediate precision was used in the calculation of expanded measurement uncertainty.

ACM: "...for all compounds (relative standard deviation, RSD<5 %)..."

**Page 10 line 306: What do you mean by assuming sampling volume? You must know this.**

AR: Yes, the sample volume is known exactly. The LOQ values were also shown in pptv units to represent ambient application and for that purpose a sample volume of 5 L was used. This sample volume would be reasonable in outdoor ambient air measurements. Obviously, as the referees' comment shows, the word "assuming" was not suitable for this and that was changed to "using". Anyone can calculate the LOQ in pptv units themselves since we provide the LOQ in absolute amounts (mass in nanograms in sorbent). The same correction was made in the Conclusions section.

ACM: In Section 3.1.4: "When converted to pptv units by using a sampling volume of 5 L..."
In Conclusions section: "If using a sample volume of 5 L in offline sampling,..."

**Page 11 line 327-9: I do not understand this sentence. Please rephrase.**

AR: This was revised. The original sentence was poorly and wrongly written. Please note that the details to these results are given in the supplement Text S2 as written at the beginning of Section 3.1.5.

ACM: "Interestingly, a substantial portion of the compound losses could be explained by the losses onto the empty tube used in online sampling, e.g. the amount of β-farnesene lost in the empty tube was ca. 44%. However, simultaneously a substantial portion of the losses could not be explained by losses onto the empty tube, e.g. up to 98% of the ent-kaurene losses could not be accounted for."

**Page 12 line 375. Please expand on why heated stainless steel better than other ozone removal mechanisms.**

AR: Discussion and references on ozone removal techniques was added

ACM: This part was added "Several different kind of ozone traps and filters have been used to trap ozone, but most of them are not suitable even for MTs and SQTs (Polmann et al. 2005, Calogirou et al. 1996). In addition, some of the ozone removal techniques have very short ozone removal capacity and they have to be changed frequently and therefore are not ideal for online sampling (Bouvier-Brown et al. 2009, Fick et al., 2001)."

The following references were added to the manuscript:

Calogirou, A., Larsen, B.R., Brussol, C., Duane, M., Kotzias, D., 1996. Decomposition of terpenes by ozone during sampling on Tenax. Analytical Chemistry 68,1499-1506.

Fick, J., Pommer, L., Andersson, B., Nilsson, C., 2001. Ozone removal in the sampling of parts per billion levels of Terpenoid compounds: an evaluation of different scrubber materials. Environmental Science and Technology 35, 1458-1462.

**Page 13 line 380. I think it would be wise and enriching the ms if you add this data.**

AR: These were preliminary experiments and performed with another GC-MS by using liquid injection. Including this preliminary screening data would not be very informative for the reader. We did not have the alkane series at the time, so even tentative identification was not attempted (only the mass spectra (m/z 272 ion specifically targeted) and the retention time expected range was evaluated for screening of the DTs). The main point in this part was to demonstrate why the birch leaves were not studied any further in the headspace experiments (no DTs could been detected in the birch solute extracts). Since the literature reference Kanerva et al. (2008) corroborated our findings, there was no apparent reason to show the data.

In principle, since this does not contribute significantly to the manuscript, all information regarding the study of birch leaves was excluded from the manuscript. Upon revision, it was considered that the information about birch leaves and DTs was not crucial, although it could have been valuable for future studies. Nonetheless, to avoid prolonging the manuscript/supplement, this information was removed.

ACM: The Section 3.2 first paragraph (starting "Prior to the dynamic headspace extraction…") of the original manuscript was removed. In addition, the mentioning of birch leaves was removed from all parts of the revised manuscript (removed from Section 2.4 and Conclusions).

**Page 13 section 3.2.1. Please explain why you heat needles and twigs.**

AR: Explanation to this has been added in the revised manuscript. This is part of the same corrections as presented earlier in these author replies.

ACM: In Section 2.4:
"…The main emphasis on real sample applications was set on the identification and study of DTs, although MTs and SQTs were also monitored. Based on previous studies, it was considered that since the DTs seem to be emitted from plant storage pools and since temperature has been shown to be a driving force for DTs emissions (Haberstroh et al., 2018;Matsunaga et al., 2012;Yáñez-Serrano et al., 2018), heating of the plant materials would be an efficient way for releasing the DTs into gas-phase. Thus, the studied plant material samples were firstly heated to temperatures ranging from 30 °C to up to 70 °C in order to obtain presumably detectable amounts of DTs to allow their identification and quantification."

In Section 3.2 the following was added:
"As explained in Section 2.4., in the following experiments the plant material samples were heated in order to characterize the DTs potentially emitted by the studied tree species and to further test the TD-GC-MS method performance in view of real sample analysis. The plant material samples were heated to facilitate the release of DTs from the plant storage pools into the gas-phase."

**Page 13 line 389. If you could only identify 5 how can you report 11?**

AR: Five out of the 14 detected compounds could be tentatively identified based on both mass spectra and RI matches. For the other nine detected compounds we proposed a chemical formula and compound group based on the mass spectra (and by sanity checking that the RI was reasonable for that formula). This is explained in Section 2.5 (see ACM). In this Section 3.2.1, a misplaced term "positively identified" was replaced with "tentatively identified". Maybe this was causing the confusion originally.

ACM: Revisions in Section 2.5:
"…If an unknown compound could not be tentatively identified (either too high RI difference or not adequately matching mass spectra), a proposed compound formula was deduced based on the mass spectra. The non-target compounds that were either tentatively identified or assigned with a formula were then categorised as MTs, SQT and DTs when applicable, the others were categorised as BVOCs. For example, if the non-target compound had the characteristic m/z 272, 257, 243 and 229 ions of DTs and the RI value was representative of DTs (Adams, 2007), the compound was assigned in this study to the plausible DT category. Similar identification procedure of non-target compounds has been used also in previous BVOC studies (e.g. Chan et al., 2016; Hellén et al., 2018; Kännaste et al., 2013; Yee et al., 2018). The compounds were marked here as *tentatively identified* since authentic standards were not available, thus the final verification is lacking…"

Revisions in Section 3.2.1:
e.g. "Of these 14 compounds, only five could be tentatively identified based on mass spectra and retention index matches."

**Page 13 line 406: what do you mean by coefficients? Please explain.**

AR: Nothing, it is there for no apparent reason.

ACM: Coefficients was removed from the sentence.

**Page 15 line 460: what do you mean by good? This is very vague, good as compared to what. Please expand.**

AR: Agreed, this was vague. The sentence was removed. In the revised version, it is replaced directly with the results.

ACM: "The sampling recovery results (15 m long Teflon FEP line, 1 m heated stainless-steel line and a Teflon cuvette bag) were mostly acceptable (within 100±20 %) for all studied compounds. For example, the recoveries of DTs were on average 86±1% and 85±6% when the 1 m heated stainless-steel and 15 m Teflon FEP sampling lines were tested, respectively. These sampling recovery results of DTs were promising, since they demonstrated that the DTs were not lost in excessive amounts in the sampling lines tested."

**Editting comments**

**Page 2 Line 34. OH must go before hydroxyl not before radical.**

AR: Corrected by assuming that the Reviewer meant after and not before. In our original version we had it "hydroxyl radical (OH) reactivity". This was changed.

ACM: Corrected to: "…total hydroxyl (OH) radical…"

**Page 2 line 35: You must explain what VOC is, you only said BVOC before.**

AR: Done.

ACM: "…reactivity has been introduced to study indirectly the volatile organic compound (VOC) content…"

**Page 7 line 219: put a before temperature and I suppose you mean that you place the temp sensor in the cuvette not the logger itself, right?**

AR: Yes, the sensor was inside the cuvette, not the logger itself. Sentence was rephrased.

ACM: "A data logger with temperature sensor was used for monitoring the cuvette inside temperature."

**Page 9 line 268: replace "with" with "for". And state which compounds you refer to.**

AR: Done by guiding the reader to see the plot.

ACM: "…(although the reproducibility was a bit poor for some compounds, see Fig. 2a)."

References:

Mermet, K., Sauvage, S., Dusanter, S., Salameh, T., Léonardis, T., Flaud, P. M., Perraudin, É., Villenave, É., and Locoge, N.: Optimization of a gas chromatographic unit for measuring biogenic volatile organic compounds in ambient air, Atmos. Meas. Tech., 12, 6153-6171, doi:10.5194/amt-12-6153-2019, 2019.

---

## Author Comment (AC2) · 3 Apr 2020

We thank the reviewer for the evaluation of the manuscript. The manuscript was revised according to the reviewers' comments. Here the point-by-point answers are given. The reviewers' comments are in bold and our answers are in normal font.  The answers are divided into author's response (AR) and author's changes to manuscript (ACM).

**Anonymous Referee #2

**The study by Helin et al. describes in depth the analytical method of a thermal desorption-gas chromatography-mass spectrometry (TD-GC-MS) to quantify a series of Biogenic Volatile Organic Compounds (BVOC). While the core analytical method has been previously used extensively, such a detailed description of the processes and parameters affecting the sampling collection is a valuable addition. The authors focus their investigations to diterpenes, a highly understudied class of compounds that exhibits difficulties in quantification due to their volatility, low abundance, sticky nature and experimental artefacts. The paper reads well but few critical points deserve more attention. Therefore, the paper is suitable for AMT and I could recommend publication but only after addressing the following comments/concerns.**

**General comments**

**1. There seems to be a confusion on the terminology that starts from the title itself and extends to various points of the text. Terpenes are a class of compounds that include monoterpenes, sesquiterpenes and diterpenes. However, it's true that in the literature terpenes and terpenoids are frequently used in the same bases. But monoterpenoids, sesquiterpenoids and diterpenoids are actually modified terpenes, with different chemical formulas (eg. Chan et al., 2016). I would recommend changing the title to the more general "Biogenic Volatile Organic Compounds" or "Terpenoids". Consequently, individual corrections on the text over case-based referral to the individual species is needed.**

AR: We agree with the Reviewer that in the literature a somewhat mixed use of terpenes and terpenoids is unfortunately common. In the revised manuscript we have changed the naming and added to all Tables the compound categories to which we refer to. This helps the reader to trace all the compounds to which we are referring to. In addition, we changed the referred title part to "…monoterpenes, sesquiterpenes and diterpenes". This was considered to being more representative of the main work.

ACM: The terminology has been changed throughout the manuscript. As an example, the compound categories were added Section 2.1. and Table 1: "The analyte included nine MTs (both hydrocarbon and oxygenated MTs), six SQTs (both hydrocarbon and oxygenated SQTs), three diterpenes (all hydrocarbon DTs) and three other compounds that were categorised here as BVOCs."

Revised Table 1:

**Table 1. List of analytes, corresponding retention indices (RI) and analytical figures of merit.**

| Compound | CAS Number | Formula | RI[a] | Quan-tify m/z ion | LOQ in pg (in pptv)[b] | Repea-tability (RSD%) | Inter-mediate precision (RSD%) | U[c] (%) | U[d] (%) |
|---|---|---|---|---|---|---|---|---|---|
| α-Pinene (MT) | 7785-70-8 | $C_{10}H_{16}$ | 945 (942) | 93 | 34 (1.2) | 2.6 | 3.8 | 16 | 18 |
| camphene (MT) | 79-92-5 | $C_{10}H_{16}$ | 963 (960) | 93 | 14 (0.5) | 1.6 | 4.0 | 16 | 18 |
| β-pinene (MT) | 19902-08-0 | $C_{10}H_{16}$ | 989 (988) | 93 | 13 (0.5) | 4.2 | 3.2 | 17 | 20 |
| 3-carene (MT) | 498-15-7 | $C_{10}H_{16}$ | 1018 (1017) | 93 | 18 (0.6) | 2.0 | 3.3 | 16 | 17 |
| *p*-cymene (MT) | 99-87-6 | $C_{10}H_{14}$ | 1033 (1031) | 119 | 20 (0.7) | 2.0 | 4.2 | 17 | 19 |
| limonene (MT) | 5989-54-8 | $C_{10}H_{16}$ | 1039 (1037) | 68 | 62 (2.2) | 1.5 | 3.8 | 17 | 17 |
| 1,8-cineol (MT) | 470-82-6 | $C_{10}H_{18}O$ | 1044 (1042) | 154 | 27 (0.9) | 2.6 | 4.7 | 18 | 20 |
| terpinolene (MT) | 586-62-9 | $C_{10}H_{16}$ | 1094 (1093) | 121 | 65 (2.3) | 2.5 | 5.0 | 17 | 18 |
| linalool (MT) | 78-70-6 | $C_{10}H_{18}O$ | 1100 (1100) | 71 | 88 (2.8) | 2.4 | 4.8 | 18 | 18 |
| 4-acetyl-1-methylcyclohexene (4-AMCH) (BVOC) | 6090-09-1 | $C_9H_{14}O$ | 1140 (1139) | 95 | 120 (4.2) | 2.7 | 6.3 | 19 | 20 |
| nopinone (BVOC) | 38651-65-9 | $C_9H_{14}O$ | 1154 (1153) | 83 | 29 (1.0) | 2.7 | 5.3 | 18 | 21 |
| bornylacetate (BVOC) | 5655-61-8 | $C_{12}H_{20}O_2$ | 1295 (1294) | 95 | 49 (1.2) | 2.4 | 6.2 | 20 | 23 |
| longicyclene (SQT) | 1137-12-8 | $C_{15}H_{24}$ | 1401 (1401) | 94 | 35 (0.8) | 2.4 | 4.9 | 18 | 21 |
| isolongifolene (SQT) | 1135-66-6 | $C_{15}H_{24}$ | 1422 (1422) | 161 | 38 (0.9) | 2.5 | 5.7 | 19 | 22 |
| β-caryophyllene (SQT) | 87-44-5 | $C_{15}H_{24}$ | 1445 (1444) | 93 | 105 (2.5) | 2.2 | 3.5 | 17 | 22 |
| β-farnesene (SQT) | 18794-84-8 | $C_{15}H_{24}$ | 1457 (1456) | 69 | 198 (4.7) | 5.0 | 5.9 | 20 | 22 |
| α-humulene (SQT) | 6753-98-6 | $C_{15}H_{24}$ | 1481 (1480) | 93 | 33 (0.8) | 2.1 | 4.2 | 18 | 24 |
| caryophyllene oxide (SQT) | 1139-30-6 | $C_{15}H_{24}O$ | 1612 (1612) | 79 | 340 (7.5) | 18.9 | 8.6 | 24 | 28 |
| cembrene (DT) | 1898-13-1 | $C_{20}H_{32}$ | 1959 (1959) | 93 | 287 (5.1) | 4.0 | 9.5 | 27 | 52 |
| ent-kaurene (DT) | 562-28-7 | $C_{20}H_{32}$ | 2106 (2106) | 257 | 518 (9.3) | 2.9 | 7.7 | 26 | 42 |
| 3-methylene-5-α-androstane (3-MA)[e] (DT) | not assigned | $C_{20}H_{32}$ | 2153 (2152) | 257 | 383 (6.9) | 3.0 | 7.2 | - | - |

[a] RI as Kovats indices (and arithmetic indices in brackets)

[b] LOQ in pptv calculated by using sample volume of 5 L (at T=298 K and p=1 atm)

[c] U (%) for ambient air application

[d] U (%) for branch enclosure application

[e] The purity of this crude standard was set to being 100% (allowing the determination of LOQ, but unknown uncertainty)

**2. My most serious concern has to do with the striking claim that there is no need for heating the Teflon lines in field applications, in contrast to almost all experimental approaches in the field to date. That conclusion was derived performing sampling recovery tests over which a second sample was taken subsequent to the original (L159: "Inlet line recovery was calculated simply by the relationship of analytes found in the second tube divided by the amounts in the first tube.") One would naturally expect that when the line is kept at the same temperature, artifacts cannot be evaluated properly. The losses on the line could be better illustrated if after the first sample, a subsequent "blank" sample over a now heated line is taken. In this approach, the amount of VOCs stuck in the tube could be quantified and the proper temperature for heating up the lines could be ascertained.**

AR: We agree with the Reviewer. The original sentence was too generalizing. It was meant that the Teflon FEP sampling line does not need to be extensively heated. It was not intended to mean that non-heated Teflon FEP lines could be used e.g. during cold spring. We revised this part so that our results better corroborate the statement and removed the field applications part from the sentence. In the revised manuscript there is no claim of non-heated lines usage in field measurements anymore.

ACM: "The results indicate that under the applied conditions the Teflon FEP sampling lines do not need to be heated excessively above room temperature (22±1 °C)"

**3. Similar to the previous comment, the authors describe in S1.5 the problems that arise with the replacement of the Teflon line. It seems that a new line is causing more artefacts comparing to a used one, indicating some saturation over the line surfaces. In general, the inlet line experiments should be explained in greater detail. The claim that Teflon lines should remain unheated for field applications has the potential to mislead future research and therefore it cannot become acceptable unless strong and well described proof of concept is presented.**

AR: The reviewer is correct. In the original manuscript we had a sentence that was too generalizing, and it could have been interpreted in a wrong way. It was not our intention to mislead the reader. The sentence was modified as shown in above point 2. ACM part.

The inlet line experiments are explained quite thoroughly in Section 2.3.3 and in the schematic illustrations of the setups presented in the supplementary (the information is presented so that others can reproduce the experiments). This was not really expanded, since upon revision, no missing information could be identified. Also, in that Section 2.3.3 another study by Hellen et al. (2012) is cited, where a similar experimental setup was used.

Regarding the Supplementary S2.5 (in the revised version it is Text S2), it is true that the line changes initially caused a drop in the overall recovery levels. The lines and connections were checked a few times to identify leaks, which were thought to be the reason, but we could not verify the leaking for sure. It was left a bit open question was the improvement overtime really caused by the "saturation" as the reviewer puts it or by the done leak checking open/closing connections. Since the leak checking information was not really scientific nor fitted the text, it was not included originally in the S2.5. However, due to this comment, it was considered to being relevant and was added to the supplement text.

Change to supplement: "It is unknown if the initially observed drop in overall recoveries was caused by some active surfaces in the Teflon line (which became saturated over time) or by potential loose connections upon line changes (connections were open/closed separately a few times to investigate possible leaking)."

**4. There is a lot of useful information, results and discussion in the supplementary information. Sometimes, it becomes challenging to follow the text while browsing through Figures, Supplementary Figures, additional discussion and references. I would recommend to bring forward the supplementary discussion, including several of the figures.**

AR: In the original manuscript, we attempted to keep the main text and figures as compact as possible and to present all complementary information in the supplementary information (SI). We wish to keep it so for the most part. However, in view of one of the Reviewer #1 comments and this comment, the supplementary figure showing the cuvette results was raised from the SI to the manuscript in order to help the reader better transit between the text and the plot for results interpretation purposes. In addition, upon revision, it was considered that the real sample chromatograms could fit in the manuscript and that these could help the reader in visualizing the results, thus those were also raised from the SI to the manuscript.

ACM: Three supplementary Figures were raised to the manuscript (now these are Figs. 3, 6 and 7). The Fig. 3 shows the cuvette recovery results, while Figs. 6 and 7 are the chromatograms from pine needles and spruce twigs extraction experiments, respectively (revised by adding blank chromatograms also to those).

**5. In order to derive some quantitative emissions of diterpenes from needles, the experiments (both in lab and field) were performed under unrealistically high temperatures. While this is acceptable for validating the method, it should be further noted that these are stored compounds coming for the pools of the plant material and further highlight the reasons for this approach. Some investigations between the vapor pressure and measured emissions would be nice addition to the paper, if possible.**

AR: Thank you for pointing this out. The information about the emissions coming from storage pools was added to justify the approach. This was added to Sections 2.4 and 3.2. (related also to Reviewer #1 comments).

In general, we agree with the Reviewer that some investigation between vapor pressure and measured emissions could be nice addition, however, we are not sure how it would fit into this manuscript and what it would contribute eventually to the overall manuscript. As an example, in Figs. AR1 and AR2 are presented the plots of emissions rates as a function of vapor pressures from the pine needles and spruce twigs experiments, respectively. The vapor pressures (in mmHg at 25 °C) were predicted values obtained via EpiSuite (https://www.epa.gov/tsca-screening-tools) and are presented only for those compounds that could be tentatively identified. The emission rates are presented at 30 °C and 60 °C as observed in the laboratory experiments performed in our study (e.g. in Fig. AR1 one of the tentatively identified diterpenes was emitted only at 60 °C, therefore the point is missing from the 30 °C results). In the plots, both x- and y-axis are presented in logarithmic scales.

As can be seen in these Figs. AR1 and AR2, the emission rates are the highest for the most volatile compounds investigated (monoterpenes) and the lowest for the least volatile compounds investigated (diterpenes). In principle, this is a fairly trivial result and does not necessarily tell anything meaningful about, for example, the terpene content of the plant storage pools. There could be also some uncertainty in the predicted vapor pressure values used here. In the end, we decided not to present these plots in the manuscript. In our opinion, the added value would have been minor and it would have not fitted into the main scope of our study.

[Figure]

Figure AR1. Relationship between emission rates (E) and vapor pressure in the emissions from pine needles laboratory experiments.

[Figure]

Figure AR2. Relationship between emission rates (E) and vapor pressure in the emissions from spruce twigs laboratory experiments.

ACM: The beginning of Section 2.4 was rewritten and modified:

"Two types of real sample applications were selected in order to test the analytical method performance, to identify compounds emitted by boreal forest tree species and to quantify the emissions from a branch enclosure. The first sample application was a dynamic headspace sampling of compounds emitted by needles/twigs in a closed chamber upon heating, and the second sample application was a study of emissions from a branch enclosure in field conditions. The main emphasis on real sample applications was set on the identification and study of DTs, although MTs and SQTs were also monitored. Based on previous studies, it was considered that since the DTs seem to be emitted from plant storage pools and since temperature has been shown to be a driving force for DTs emissions (Haberstroh et al., 2018;Matsunaga et al., 2012;Yáñez-Serrano et al., 2018), heating of the plant materials would be an efficient way for releasing the DTs into gas-phase. Thus, the studied plant material samples were firstly heated to temperatures ranging from 30 °C to up to 70 °C in order to obtain presumably detectable amounts of DTs to allow their identification and quantification."

In the second paragraph of Section 3.2 the following was added:

"As explained in Section 2.4., in the following experiments the plant material samples were heated in order to characterize the DTs potentially emitted by the studied tree species and to further test the TD-GC-MS method performance in view of real sample analysis. The plant material samples were heated to facilitate the release of DTs from the plant storage pools into the gas-phase."

**Specific comments**

**L34-40. The authors probably refer to the comparative reactivity method. If so, the reference of Kovacs and Brune is misplaced. Instead the study by Sinha et al (2008) should be cited.**

AR: Yes, it should be Sinha et al. (2008) not Kovacs and Brune.

ACM: Reference Kovacs and Brune removed and was changed to Sinha et al. (2008).

**L93. Please make a comment on how the commercial adsorbent tubes compare with the custom-made ones.**

AR: In this work, the commercial and custom-made tubes were used randomly. For example, the intermediate precision (n=22) and blanks (n=10) used for LOQ determination contain results from both types of tubes. This information was added.

ACM: "The commercial and in-house prepared tubes were used collectively and randomly in this work."

**L154-155. Zero air generators and carbon cartridges have frequently shown that are not completely scrubbing the VOCs. Have you tested the efficiency of these methods? How do they compare between them?**

AR: In our case both purification systems worked fine for our purpose, at least they scrubbed the BVOCs we were investigating. In this work, we did not compare the different purification setups.

This is said in the manuscript in Section 2.5: "In both real sample applications, the $C_{in}$ was negligible (or <LOQ) for all compounds, and was subsequently set to zero in the calculations."

**L182&L239. Please use equations in separate lines so they are clearly distinguishable from the rest of the text.**

AR: Done.

ACM: Equations were added to separate lines.

**L375. O3 removal. Have you tested other scrubbers? Why the heated stainless steel in the optimal solution of ozone removal and not e.g. KI scrubbers or Na2S2O3 infused quartz filters? Please refrain from strong claims that are not supported by experiments.**

AR: Discussion and references on ozone removal techniques was added. Changes were made to avoid too strong claims, e.g. the "optimal solution" was replaced with "good solution".

ACM: This part was added "Several different kind of ozone traps and filters have been used to trap ozone, but most of them are not suitable even for MTs and SQTs (Polmann et al. 2005, Calogirou et al. 1996). In addition, some of the ozone removal techniques have very short ozone removal capacity and they have to be changed frequently and therefore are not ideal for online sampling (Bouvier-Brown et al. 2009, Fick et al., 2001)."

In addition the sentence was modified a bit "Based on the ozone removal results presented in Hellén et al. (2012) and the inlet recovery results presented in this study (Fig. 2d), heated stainless-steel line could be a good solution for ozone removal especially in online sampling." to avoid too strong claims.

The following were added to References:

Calogirou, A., Larsen, B.R., Brussol, C., Duane, M., Kotzias, D., 1996. Decomposition of terpenes by ozone during sampling on Tenax. Analytical Chemistry 68,1499-1506.

Fick, J., Pommer, L., Andersson, B., Nilsson, C., 2001. Ozone removal in the sampling of parts per billion levels of Terpenoid compounds: an evaluation of different scrubber materials. Environmental Science and Technology 35, 1458-1462.

**L386. Please make a comment on the saturated peaks of Sup.Fig.19-20.**

AR: Those are not saturated peaks, the chromatograms are just scaled in such a way that all the peaks are visible. The comment clarifying this is present in the respective captions, e.g.: "The chromatogram is scaled for presentation purposes, peak number 3 is not showed in full height in order to visualize the other peaks more clearly." These Figures are now in the manuscript, Figs. 6 and 7. No changes were made regarding this since the information is in the captions.

**L463. Similar to my major comments, it has not been sufficiently demonstrated that the ambient sampling DTs does not require heating lines. One potential experiment to help towards this direction would be to sample through a long inlet line the analyte mixture at different temperatures and present a plot of inlet temperature vs signal.**

AR: This sentence was removed and the paragraph was revised. The last sentence of the paragraph was changed.

ACM: "These sampling recovery results of DTs were promising, since they demonstrated that the DTs were not lost in excessive amounts in the sampling lines tested."

**References Chan, A. W. H., Kreisberg, N. M., Hohaus, T., Campuzano-Jost, P., Zhao, Y., Day, D. A., Kaser, L., Karl, T., Hansel, A., Teng, A. P., Ruehl, C. R., Sueper, D. T., Jayne, J. T., Worsnop, D. R., Jimenez, J. L., Hering, S. V., and Goldstein, A. H.: Speciated measurements of semivolatile and intermediate volatility organic compounds (S/IVOCs) in a pine forest during BEACHON-RoMBAS 2011, Atmos. Chem. Phys., 16, 1187–1205, https://doi.org/10.5194/acp-16-1187-2016, 2016.**

Sinha, V., Williams, J., Crowley, J. N., and Lelieveld, J.: The Comparative Reactivity Method – a new tool to measure total OH Reactivity in ambient air, Atmos. Chem. Phys., 8, 2213–2227, https://doi.org/10.5194/acp-8-2213-2008, 2008.

---

## Referee Report (RR1)

The paper is well written and can be accepted after taking into consideration the following revisions :

1- I suggest to replace the word « development » in the paper's title by « optimization » : because the technique has already been developed, and in this paper it was optimized.
2- Uncertainty estimation : based on « ACTRIS 2018, deliverable 3.17 », the uncertainty related to the peak integration should be addressed ; as well as the one related to the memory effect.
3- Line 445 : please reconsider what you added in the paragraph regarding ozone trap, with regards to the work performed by Mermet et al. 2019 where they thoroghly investigated many types of ozone scrubbers for the online measurement of MT and SQT by TD-GC-FID.

---

## Author Response (AR2)

We thank the reviewers for taking the time to evaluate the manuscript. The manuscript was revised according to the reviewers' comments. Here the point-by-point answers are given. The reviewers' comments are in bold and our answers are in normal font. The answers are divided into author's response (AR) and author's changes (AC).

**The paper is well written and can be accepted after taking into consideration the following revisions:**

1- **I suggest to replace the word « development » in the paper's title by « optimization » : because the technique has already been developed, and in this paper it was optimized.**

A: Changed as suggested, but used "optimisation" for consistent spelling according to the journal guidelines.

AC: Changed title to "Optimisation of a thermal desorption-gas chromatography-mass spectrometry method for the analysis of monoterpenoids, sesquiterpenoids and diterpenoids"

2- **Uncertainty estimation : based on « ACTRIS 2018, deliverable 3.17 », the uncertainty related to the peak integration should be addressed ; as well as the one related to the memory effect.**

A: The peak integration uncertainty was assessed in the revised manuscript and the description of the process has been added to supplement. The corresponding U(%) were updated.

In our case, the uncertainty related to peak integration was fairly minor, since the peak shapes were mostly good (e.g. no significant tailing in individual peaks and no overlapping peaks) and because the selected-ion recording (SIR) used in MS detection facilitated the use of distinct peaks. In ACTRIS (2018) guidelines, it is mentioned that the peak uncertainty is needed mainly for peaks strongly deviating from the Gaussian peak form.

The uncertainty related to memory effect (or carryover) was not considered here since the desorption efficiency was ≥99.8 for all compounds. Thus, the uncertainty related to memory effect was considered to being negligible.

AC: In supplement:

"The combined standard uncertainty (total uncertainty, $u\mathcal{X}_{\text{total}}$) includes both the random and systematic errors affecting the measurements. In this work, the $u\mathcal{X}_{\text{total}}$ was calculated for both procedures i) and ii) by using Eq. (S1):

$$u\mathcal{X}_{\text{total}}{}^2 = u\mathcal{X}_{\text{prec}}{}^2 + u\mathcal{X}_{\text{stdprep}}{}^2 + u\mathcal{X}_{\text{vol}}{}^2 + u\mathcal{X}_{\text{int}}{}^2 + u\mathcal{X}_{\text{rec}}{}^2 \ , \tag{S1}$$

where $u\mathcal{X}_{\text{prec}}$ is the precision component taking into account random errors, $u\mathcal{X}_{\text{stdprep}}$ is the uncertainty due to calibration standard preparation, $u\mathcal{X}_{\text{vol}}$ is the component descriptive of systematic errors in the sample volume determination, $u\mathcal{X}_{\text{int}}$ is the systematic integration error, and $u\mathcal{X}_{\text{rec}}$ is the recovery component taking into account the losses in sampling lines and/or cuvette."

"The $u\mathcal{X}_{\text{int}}$ was obtained by integrating the peak areas in such a way that a definite overestimation of the peak area ($A_{max}$) and underestimation of the peak area ($A_{min}$) were made, similar as shown in ACTRIS, 2018. The potential error in peak area integration ($uA_{\text{int}}$) was then obtained by using Eq. (S3), which gives a conservative estimate of the error (ACTRIS, 2018). This was then converted to respective analyte concentrations to obtain $u\mathcal{X}_{\text{int}}$.

$$uA_{\text{int}} = \frac{A_{max} - A_{min}}{2} \tag{S3}"$$

In the main manuscript, the re-calculated expanded measurement uncertainty (U) values were updated in all corresponding places in the text and in Table 1.

3- **Line 445 : please reconsider what you added in the paragraph regarding ozone trap, with regards to the work performed by Mermet et al. 2019 where they thoroghly investigated many types of ozone scrubbers for the online measurement of MT and SQT by TD-GC-FID.**

A: Most likely the Reviewer refers to the manuscript paragraph starting in Line 421. The work performed by Mermet et al. (2019) was added as a reference in the paragraph and a notation to it was separately made.

[revised manuscript text omitted]